# Substantial differences in source contributions to carbon emissions and health damage necessitate balanced synergistic control plans in China

Yilin Chen [1,2] ✉, Huizhong Shen [2,3], Guofeng Shen [4], Jianmin Ma [4], Yafang Cheng [5], Armistead G. Russell [6], Shunliu Zhao[7], Amir Hakami [7] & Shu Tao [2,4]

China's strategy to concurrently address climate change and air pollution mitigation is hindered by a lack of comprehensive information on source contributions to health damage and carbon emissions. Here we show notable discrepancies between source contributions to $CO_2$ emissions and fine particulate matter ($PM_{2.5}$)-related mortality by using adjoint emission sensitivity modeling to attribute premature mortality in 2017 to 53 sector and fuel/process combinations with high spatial resolution. Our findings reveal that monetized $PM_{2.5}$ health damage exceeds climate impacts in over half of the analyzed subsectors. In addition to coal-fired energy generators and industrial boilers, the combined health and climate costs from energy-intensive processes, diesel-powered vehicles, domestic coal combustion, and agricultural activities exceed 100 billion US dollars, with health-related costs predominating. This research highlights the critical need to integrate the social costs of health damage with climate impacts to develop more balanced mitigation strategies toward these dual goals, particularly during fuel transition and industrial structure upgrading.

China is facing dual challenges related to air pollution and climate change[1–3]. To address the rising challenge of climate change, the government of China has pledged to reach its peak in carbon emissions before 2030 and achieve carbon neutrality by 2060[4]. Despite substantial improvements over the past 30 years, China is still actively mitigating severe air pollution, which causes 1.4 million premature deaths annually[5]. Notably, around 80% of these deaths are linked to

ambient particulate matter with a diameter of 2.5 micrometers or smaller ($PM_{2.5}$), which exacerbates mortality risks associated with ischemic heart disease, cerebrovascular disease, chronic obstructive pulmonary disease, lung cancer, and lower respiratory infections[6,7]. China is currently prioritizing synergistic control to address both issues because they share common sources (e.g., fossil fuel combustion)[4,8]. Measures, including improving energy efficiency,

[1]School of Urban Planning and Design, Peking University Shenzhen Graduate School, Shenzhen 518055, China. [2]Shenzhen Key Laboratory of Precision Measurement and Early Warning Technology for Urban Environmental Health Risks, School of Environmental Science and Engineering, Southern University of Science and Technology, Shenzhen 518055, China. [3]Guangdong Provincial Observation and Research Station for Coastal Atmosphere and Climate of the Greater Bay Area, School of Environmental Science and Engineering, Southern University of Science and Technology, Shenzhen 518055, China. [4]College of Urban and Environmental Sciences, Peking University, Beijing 100871, China. [5]Max Planck Institute for Chemistry, Mainz 55128, Germany. [6]School of Civil and Environmental Engineering, Georgia Institute of Technology, Atlanta, GA 30332, USA. [7]Department of Civil and Environmental Engineering, Carleton University, Ottawa, ON K1S5B6, Canada. ✉e-mail: ylchen2023@pku.edu.cn

adjusting industry structures, and increasing electrification from renewable energy, can all reduce air pollutant emissions and achieve health co-benefits.

Studies have shown that the implementation of climate policies can help improve air quality and reduce associated health losses[9,10]. The control over air pollution can also boost decarbonization[11]. For example, researchers have estimated that $CO_2$ emission reductions following the Representative Concentration Pathways 4.5 scenario (RCP4.5, which depicts a moderately warming future) can help prevent 1.3 million premature deaths globally by 2050 by reducing air pollutant emissions simultaneously[9]. This number could double under scenarios in which more stringent $CO_2$ emission control is applied[12]. However, climate policies may not adequately address air pollution-driven health losses, especially under unequal low-carbon development conditions that overlook less-developed regions[13]. Despite the promising air quality and health co-benefits attained from implementing climate policies, recent scenario analyses suggest that China cannot meet the World Health Organization air quality guidelines by implementing only climate policies[11,14].

Coordinated solutions to dual challenges must be informed by location- and sector-specific health co-benefits. The extent of co-benefits should vary substantially depending on the proximity of sources to densely populated regions, source strength, end-of-pipe technology applied, and atmospheric conditions[15,16]. However, the location-specific source attribution of national health co-benefits is a key knowledge gap in scenario-based analyses, which quantify co-benefits only at aggregated levels[11,14]. Recent finer-resolution studies have revealed substantial spatial variation in health co-benefits to motivate spatially nuanced mitigation measures. However, limited by the high computational cost required for high-resolution analyses, these studies have focused only on limited sectors[16–18].

In this study, we perform a detailed quantitative analysis to delineate the contributions of diverse sectors and their associated fuel/process combinations to $CO_2$ emissions and the health damage related to air pollution in China. This analysis encompasses critical sectors such as energy production, industry, transportation, domestic activities (both residential and commercial sources), and agriculture. The health impact assessment focuses on nationwide premature mortality attributable to ambient $PM_{2.5}$ exposure. Our coordinated source attribution analysis integrates coupled emission inventories for both air pollutants and $CO_2$ with an advanced adjoint tool developed for a regional air quality model, the Community Multiscale Air Quality (CMAQ) model. This tool allows for the efficient calculation of backward sensitivities, thereby quantifying the impact of high-resolution emission changes on nationwide premature mortality counts. The analysis includes impact of emission from speciated primary $PM_{2.5}$, which include organic carbon (OC), elementary carbon (EC), and other primary $PM_{2.5}$ particles, as well as from principal precursors of secondary $PM_{2.5}$, which include sulfur dioxide ($SO_2$), nitrogen oxides ($NO_x$), and ammonia ($NH_3$). Moreover, we propose using the integrated costs – representing the aggregate of monetized social costs from health damage and climate change – as a comprehensive metric for prioritizing sectoral and spatial control targets.

## Results

### Substantial disparities in sectoral contributions to health damage and $CO_2$ emissions

The source contribution of health-threatening air pollutants differs from that of $CO_2$, although both mainly originate from fossil fuel combustion. An analysis of the coordinated emission inventories for $CO_2$ and air pollutants for the year 2017 reveals that 86% of the total anthropogenic $CO_2$ emissions were predominantly from the energy generation and industrial sectors. Notably, coal combustion was the major contributor, accounting for 97% and 50% of the total $CO_2$ emissions in these two sectors, respectively (Fig. 1a). However, as a result of implementing multiple pre- and post-combustion control technologies[19], coal combustion in these sectors contributed 17% of the primary $PM_{2.5}$ emissions (Supplementary Fig. 1). Coal and solid-biomass combustion in the domestic sector contributed 38% of the total $PM_{2.5}$ emissions and 22% of the total $SO_2$ emissions. This substantial contribution is attributed to the restricted air supply, poor mixing, and insufficient emission control in household stoves, coupled with the use of low-quality fuels[20]. Adjoint-based source attribution indicates that more than a quarter of premature deaths attributable to ambient $PM_{2.5}$ exposure originate from the domestic sector, despite its mere 4% contribution to $CO_2$ emissions (Fig. 1b). In contrast, while the energy generation sector is an important contributor to $CO_2$ emissions, it accounts for only 10% of premature deaths. The finding that the domestic sector is the leading contributor to health losses attributable to ambient $PM_{2.5}$ exposure is consistent with findings in previous studies[21,22]. Certain subsectors contribute to health losses without being directly linked to $CO_2$ emissions from fuel combustion. These include non-combustion industrial processes, agricultural fertilizer application and livestock management, and crop residue burning in the domestic sector (Supplementary Table 1). While the combustion of crop residues does emit $CO_2$, this subsector is considered carbon neutral based on the assumption that the released $CO_2$ is later reabsorbed through subsequent biomass regrowth[23]. As in our approach for other fuel types, life cycle emissions from biofuel processing and land use changes are not included[24,25]. We term these subsectors "non-synergistic subsectors", which account for 23% of the total premature deaths attributable to the five sectors assessed.

Seven subsectors were identified as significant contributors to health damage, each of which was attributable for more than 100,000 premature deaths (Fig. 1c). These subsectors include bituminous coal combustion in the domestic, energy generation, and industrial sectors; emissions from diesel-powered vehicles; and activities in hydraulic cement production, iron and steel production, as well as agricultural fertilizer application and livestock management. While these subsectors comparably contribute to health damage, their contributions to $CO_2$ emissions differ substantially. For instance, agricultural activities significantly contribute to $PM_{2.5}$-related health damage from $NH_3$ emissions without being major contributors to $CO_2$ emissions directly. This discrepancy is also observed in the analysis of $CO_2$ emissions from other subsectors. For example, the $CO_2$ emissions from domestic bituminous coal combustion are 52–93% lower than those from the remaining five subsectors. In terms of the speciated emission contribution to health damage, more than half of the health damage from hydraulic cement and iron production, as well as domestic bituminous coal combustion, is attributed to primary $PM_{2.5}$ emissions. In contrast, health damage from bituminous coal combustion in the energy generation and industrial sectors, as well as from diesel vehicles, is predominantly attributable to $SO_2$ and $NO_x$ emissions (Supplementary Fig. 2). The impact of various fuel types and industrial processes on premature deaths within a sector also shows substantial disparities. For example, diesel-powered vehicles, which are responsible for 210,000 premature deaths, contribute eightfold more to health damage than do gasoline-powered vehicles, which emphasizes the great importance of diesel vehicle electrification in reducing air pollution and associated health risks. Overall, the ratio of source contributions to health damage versus to $CO_2$ emissions varies extensively across subsectors, ranging from 0.094 in the industrial combustion of dry natural gas to 22 in the domestic combustion of unorganized waste. This noteworthy variation in source profiles for health damage and $CO_2$ emissions underscores that strategies aimed at reducing air pollution-related health damage may not always align with those targeting decarbonization. This necessitates the formulation of coordinated, synergistic control plans to effectively balance the two objectives.

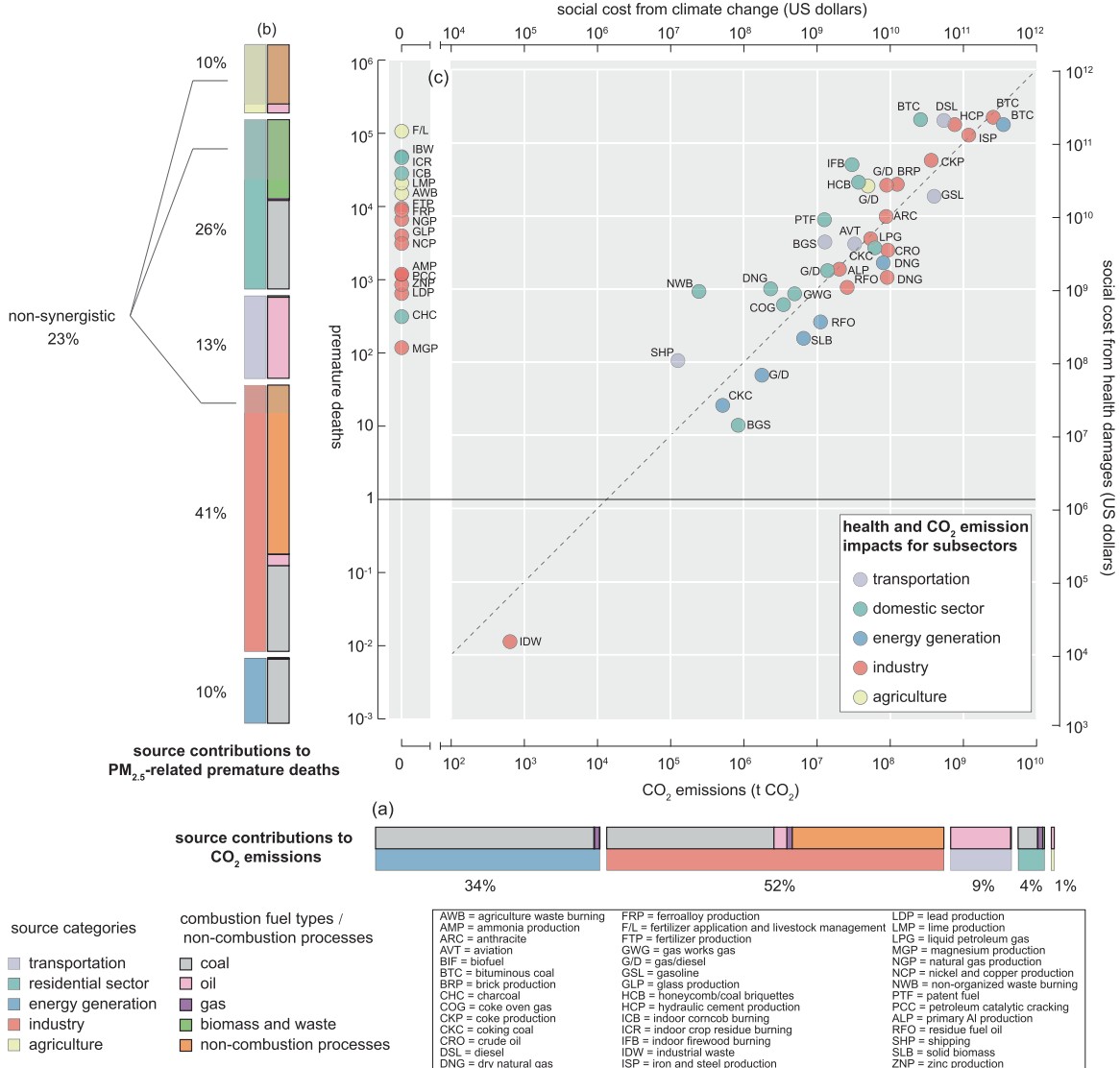

**Fig. 1 | Disparities in source attributions of CO₂ emissions and health damage.** **a** Source contributions to $CO_2$ emissions in China in 2017 based on the coordinated $CO_2$ emission inventory with the air pollutants. The lower bar represents the contributions of different sectors, and the upper bar represents the contributions of different fuel/process categories. **b** Source contributions to premature deaths attributable to long-term ambient $PM_{2.5}$ exposure in China in 2017 based on the Adjoint simulation of the Community Multiscale Air Quality modeling. The left bar represents the contributions of different sectors, and the right bar represents the contributions of different fuel/process categories. **c** Comparison of impacts on health damage and $CO_2$ emissions for each subsector. The corresponding monetized impacts are also shown on the minor axis based on a uniform statistical life value (1.33 million US dollars per premature death) and social cost of carbon (100 US dollars per ton of $CO_2$ emission). Source data are provided as a Source Data file.

## Spatial heterogeneity in contributions to health damage and CO₂ emissions

The integration of high-resolution health damage attribution data with the coordinate $CO_2$ emission inventory also enables the delineation of spatial heterogeneities in contributions to health damage and $CO_2$ emissions. By calculating the ratio of the percentage contributions to the nationwide $PM_{2.5}$-related health damage versus to $CO_2$ emissions at the gridded or regional levels, we can assess the relative significance of the contributions. In densely populated areas (Supplementary Fig. 3), including the eastern and central regions and the Sichuan Basin, the contribution to health damage notably surpassed that to $CO_2$ emissions (Fig. 2a). The highest ratio was observed in Hubei (3.9), followed by Henan and Chongqing (3.8 and 3.7, respectively), whereas the lowest ratio was recorded in Xizang (0.11). Spatial heterogeneity in the source contributions to health damage and $CO_2$ emissions is also evident within individual sectors. For instance, in the energy generation sector, Inner Mongolia contributes marginally more to $CO_2$ emissions

(3.8%) than Shandong (3.5%), yet its impact on health damage (0.75%) is significantly lower than Shandong's (1.3%) (Fig. 2b). Similarly, in the industrial sector, Guangdong's $CO_2$ emissions surpass those of Anhui and Hebei (18%), yet its health damage contribution is merely half of that observed in the latter provinces. This spatial variability is influenced not only by population density but also by the intensity of air pollutant emissions (Supplementary Fig. 4). For example, Sichuan, with its lower coal quality leading to higher emissions of $PM_{2.5}$ and $SO_2$ per ton of $CO_2$ emission, exhibits the most pronounced disparity in the energy generation sector. Similarly, the elevated $PM_{2.5}$ emissions associated with clinker production in the hydraulic cement manufacturing make Anhui and Hebei the regions with the highest discrepancy in the industrial sector.

The spatial distribution of the health damage to $CO_2$ emissions ratio is significantly influenced by population density. A log-linear regression analysis at the city level shows positive correlation between population density and the ratio of each city's percentage contribution

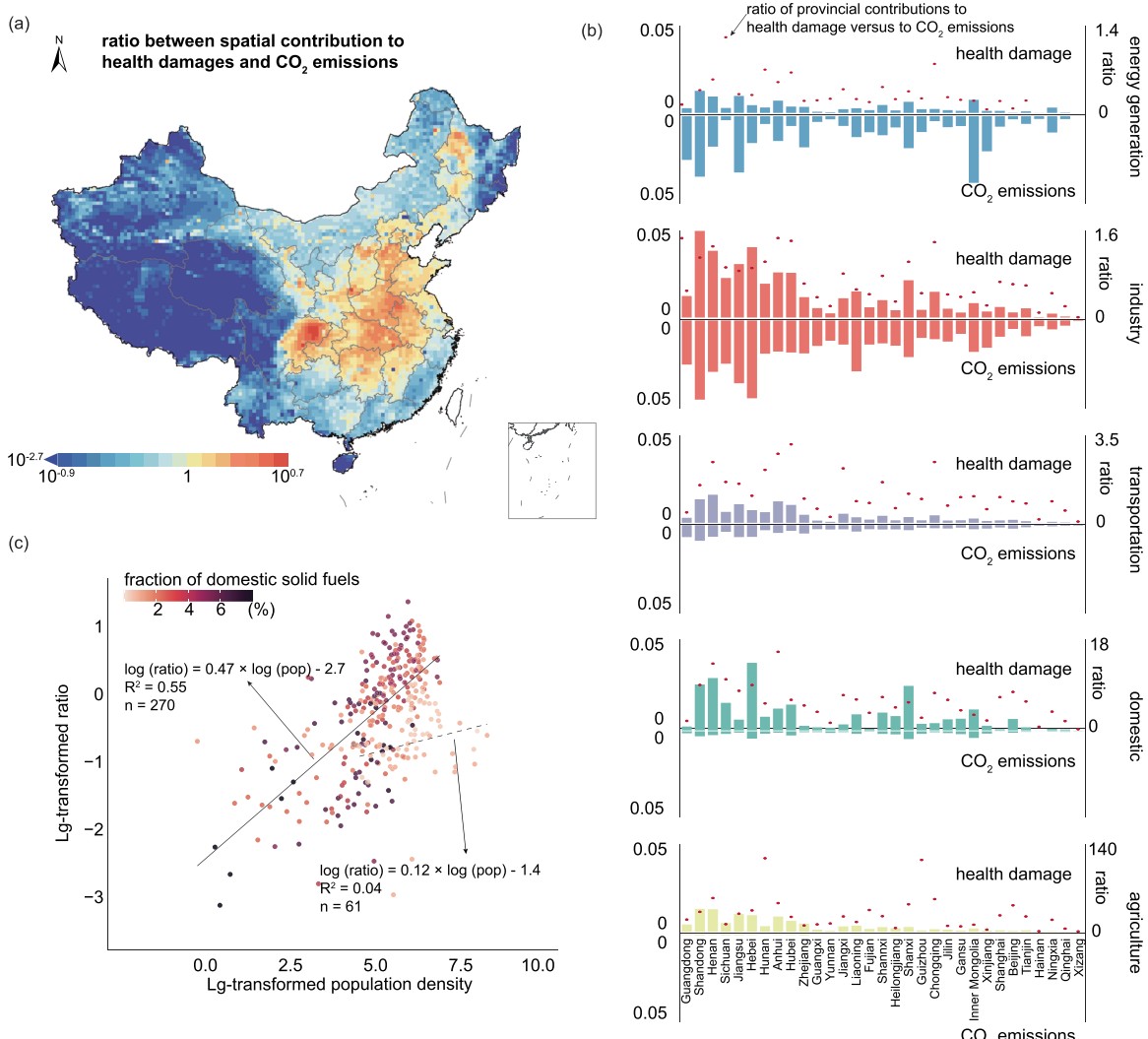

**Fig. 2 | Disparities in spatial distributions of $CO_2$ emissions and health damage.** **a** Ratio between the gridded contribution to $PM_{2.5}$ exposure-related health damage and to $CO_2$ emissions. The provincial boundary shapefile is obtained from Harvard Dataverse (https://doi.org/10.7910/DVN/DBJ3BX) and is publicly available under the Creative Commons CC0 Public Domain Dedication. **b** Comparison of provincial contributions to health damage and $CO_2$ emissions in each sector. The red dots represent the ratio of provincial contributions to health damage versus to $CO_2$ emissions. The provinces are ranked from large to small in population size. **c** Correlation between population density (pop) and the ratio of each city's contributions to health damage versus to $CO_2$ emissions (ratio). The dot color represents the domestic solid fuel usage fraction, which is defined as the fraction of energy consumed as solid fuels in the domestic sector out of the total province-level energy consumption. The solid line and dashed line represent the correlations for cities with domestic solid fuel consumption above and less than 1%, respectively, of the total provincial energy consumption. Source data are provided as a Source Data file.

to nationwide $PM_{2.5}$-related health damage versus its contribution to $CO_2$ emissions (Fig. 2c). These findings are consistent with previous studies showing that the regional health impact of ambient air pollution and the effectiveness of mitigation measures are more important in densely populated regions[26,27]. Additionally, the source profile, particularly the domestic energy structure, contributes to the spatial variations in the ratio. The color-coded scatter plot in Fig. 2c illustrates the impact of reliance on solid fuels for domestic energy needs on the correlation between the ratio of contributions to health damage versus that to $CO_2$ emissions and population density. Notably, this correlation intensifies in cities with a greater dependence on solid fuels for domestic energy, explaining more than half of the variance in city-level ratios. Specifically, a 1% increase in population density correlates with a 0.47% increase in the ratio. Conversely, in cities where domestic solid fuel consumption constitutes less than 1% of total provincial energy usage, population density accounts for merely 4% of the ratio variance, with a 1% population increase leading to a modest 0.12% increase in the ratio. The higher ratio in Beijing than in other megacities, including

Shanghai, Guangzhou, and Shenzhen, further demonstrates the critical role of domestic solid fuel consumption (Supplementary Fig. 5). Although Beijing has already decommissioned its energy-intensive industries and coal-fired power plants, it is the only city among the four megacities whose contribution to health damage exceeds its contribution to $CO_2$ emissions. This increase in Beijing's contribution to health damage was primarily attributed to the consumption of 1.8 Mt of raw coal by rural residents in 2017[28].

## Monetized social costs of $CO_2$ emissions and $PM_{2.5}$ exposure-related health damage

We further monetized health damage and climate impacts by applying a uniform value of statistical life (VSL) of 1.33 million US dollars per statistical death and a direct social cost of carbon (SCC) of 100 US dollars per ton of $CO_2$ emission (Methods). The monetized costs for health damage and climate impacts attributable to different subsectors are depicted along the minor axis in Fig. 1c. For the 36 synergistic subsectors, the monetized health damage and $CO_2$-related

climate impacts fall along the one-by-one line on a logarithmic scale, with the monetized health damage exceeding the corresponding monetized climate impacts for half of the subsectors. This finding indicates that the social costs due to climate change are comparable to the monetized concurrent health losses attributable to air pollution. The VSL value employed in this study, following the Organisation for Economic Co-operation and Development (OECD) recommended benchmark, is 2–3 times lower than the estimates recommended by the United States Environmental Protection Agency (USEPA). If higher VSL estimates were used (Supplementary Fig. 6), monetized health damage would surpass climate impacts for the majority of subsectors. This alignment, or in certain cases, the excess of monetized health damage related to climate impacts, emphasizes the near-term health benefits as an important incentive for $CO_2$ emission reduction[12,29] and justifies more ambitious decarbonization plans.

We further conducted a comparison between the monetized health damage and climate impacts with sectoral gross domestic product (GDP), assigning production-based emissions to 42 economic sectors by using the China multi-regional input-output model table for 2017[30] (Supplementary Note 1, Supplementary Table 2). On average, the integrated costs of the 42 economic sectors equal 20% of the total GDP. For most of the economic sectors, the integrated costs are lower than 5% of the sectoral GDP (Supplementary Fig. 7). However, in four sectors, the integrated costs exceed the sectoral GDP. These sectors include the production and distribution of electric power and heat power; the manufacture of non-metallic mineral products, the processing of petroleum, coking, and nuclear fuel; and the smelting and processing of metals. This comparison further underscores the critical need for decarbonizing electricity generators and energy-intensive industrial processes.

## Integrating costs from health damage modifies control priorities

Hence, we suggest employing a unified indicator, calculated as the sum of social costs from $CO_2$-related climate change and $PM_{2.5}$ exposure-related health damage. This indicator is proposed for harmonizing sectoral and spatial control objectives in co-optimal mitigation strategies. The model estimates that in 2017, the integrated costs for seven subsectors exceeded 100 billion US dollars. The highest integrated costs were associated with bituminous coal combustion in the energy generation sector, followed by industrial consumption of bituminous coal, diesel-powered vehicles, hydraulic cement production, iron and steel production, domestic consumption of bituminous coal, and agricultural fertilizer application and livestock management (Supplementary Table 1). Except for the last non-synergistic subsector, the other six subsectors all heavily relied on coal or diesel fuel. The social cost from climate change accounted for more than two-thirds of the integrated benefits from bituminous coal combustion in the energy generation sector, while the social cost from health damage accounted for 47%–89% of the integrated costs in the other five synergistic subsectors. Mitigation measures oriented toward air pollution control have been applied in these five subsectors, including household fuel switching and after-treatment technologies adopted for reducing exhaust emissions[31,32]. Their important contribution to the integrated costs encourages the implementation of synergistic control measures to mitigate $CO_2$ and air pollutant emissions. Because of their substantial contribution to health damage, the ranking position of domestic subsectors reliant on solid fuels increased in the integrated cost-based ranking (Supplementary Table 1). In addition to the domestic bituminous coal combustion, indoor crop residue burning and brush wood burning were positioned 9th and 10th, respectively, out of the 53 subsectors assessed in terms of their contribution to integrated costs. However, decarbonization-oriented mitigation measures may not address these two subsectors due to their negligible contribution to $CO_2$ emissions.

Spatially, the distributions of social costs from health damage and climate change, as well as the integrated costs, were uneven (Fig. 3c). The distributions of the contributions of climate change to social costs are skewed toward less populated regions, while the distributions of social costs from health damage are skewed toward densely populated regions (Fig. 3d). The incorporation of health co-benefits amplified the contribution of high-population regions to integrated benefits because of the proximity of air pollution and exposed populations. The areas with high integrated costs were concentrated in populated areas where greater amounts of air pollutants and $CO_2$ emissions were emitted from fuel combustion to support intensive human activities. The grids that contributed more than 1 billion US dollars in integrated costs covered only 11% of the land but hosted 60% of the population of China. Spatial analysis at $36 \times 36$ km resolution identified densely populated cities as the primary contributors to integrated costs, including Chongqing, Zhengzhou, Shanghai, Wuhan, and six cities in the Beijing, Tianjin, and Hebei (BTH) city cluster. Among these populated cities, monetized health costs typically comprised 51%–84% of the integrated costs.

Consequently, the spatial priorities for synergistic control identified based on the integrated costs differ from those based solely on the social cost of $CO_2$-related climate change. In addition to the emission amount, the integrated costs depend on the population density, energy and sector structure, and atmospheric conditions. Regarding provinces, the rankings based on the social costs from climate change were high in heavy industrial and energy-supplying provinces, including Jiangsu, Guangdong, and Inner Mongolia (Fig. 4). Because of the lower contribution to social costs from health damage, the rankings for these provinces decreased when they were sorted by integrated costs. The provincial contributions to social costs from health damage were lower in Jiangsu and Guangdong Provinces, China's most developed industrial hubs, because these provinces have made greater efforts to upgrade their industrial structure to curb emissions[33]. In Inner Mongolia, the contribution to health damage is lower because of the lower population density, despite the heavy reliance on coal. In contrast, because of the disproportionate regional contribution to health damage in densely populated areas, the rankings based on integrated costs prioritized inland provinces with high population densities. The gap between the rankings based on the integrated costs and social costs from climate change is the largest in Chongqing, one of the most densely populated cities.

In six provinces, namely, Henan, Liaoning, Tianjin, Beijing, Hebei, and Shanxi, the primary subsector contributing to integrated costs diverges from the leading subsector contributing to social costs from $CO_2$-related climate change. In Henan and Liaoning, the predominant subsector shifts from being bituminous coal combustion in the energy generation sector to being bituminous coal combustion in industrial boilers, and in Tianjin, the predominant sector changes to iron and steel production. Although the energy generation sector is the most studied sector for low-carbon pathways[16,34], the combined integrated costs from bituminous coal combustion and five energy-intensive processes (iron and steel, coke, brick, hydraulic cement, and lime production) dominated in 26 out of the 31 provinces. Thus, programs to decarbonize industrial boilers and phase out energy-intensive and highly polluting industrial processes are the keys to optimizing low-carbon development in most provinces. For Beijing, Hebei, and Shanxi, the top contributing subsectors shift to domestic bituminous coal combustion, a shift attributed to its substantial contribution to health damage.

## Discussion

The substantial contribution of monetized health damage to integrated costs suggests that sectoral control priorities for addressing air pollution may not align with the primary targets of decarbonization plans. Notably, the combustion of bituminous coal for energy

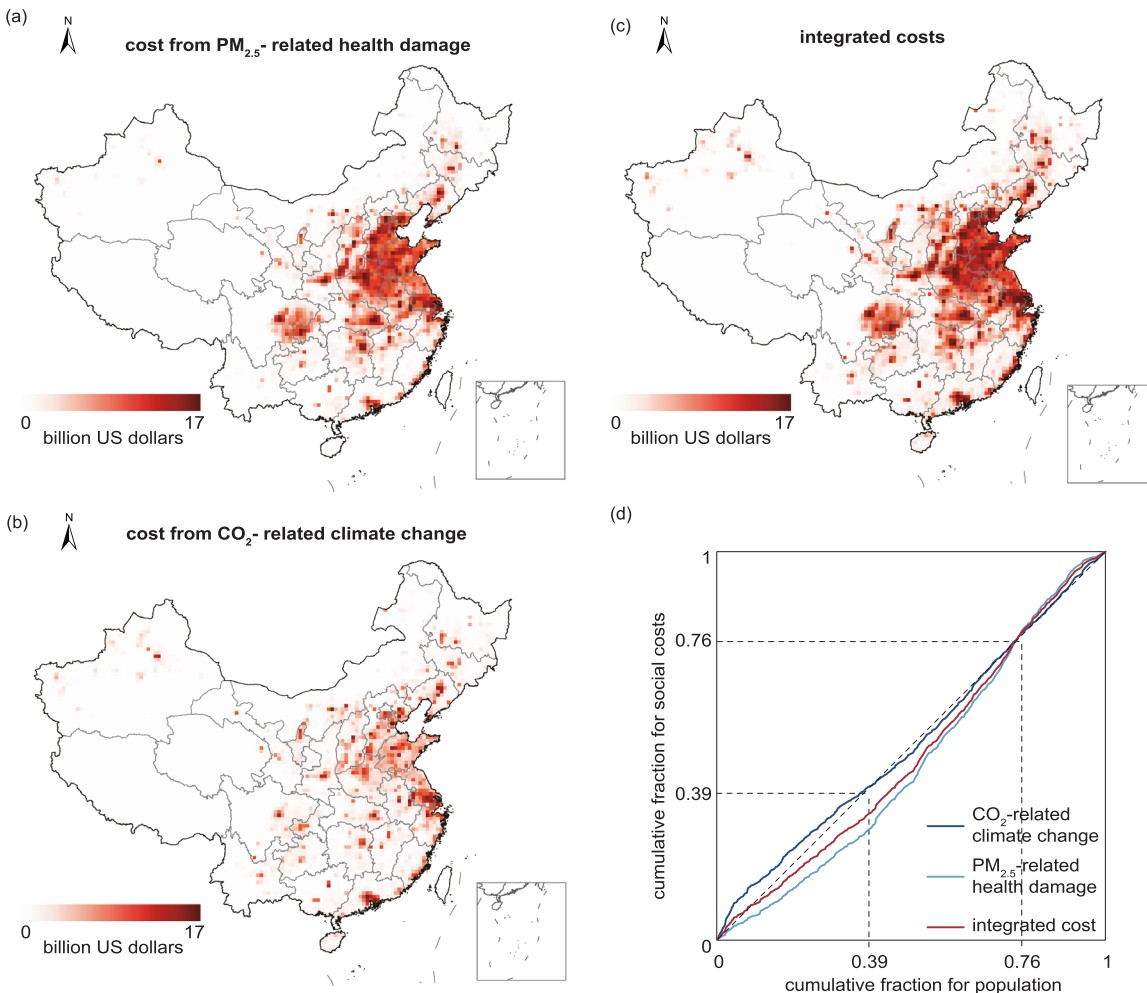

**Fig. 3 | Spatial distribution of social costs from PM$_{2.5}$ exposure-related health damage, CO$_2$-related climate change, and integrated costs. a** Spatial distribution of PM$_{2.5}$ exposure-related health damage in China at a 36 km by 36 km resolution in 2017. **b** Spatial distribution of social costs from CO$_2$-related climate change. **c** Spatial distribution of the integrated costs, which equal the sum of the monetized health damage and social cost from CO$_2$-related climate change. The provincial boundary shapefile used in a-c is obtained from Harvard Dataverse (https://doi.org/10.7910/DVN/DBJ3BX) and is publicly available under the Creative Commons CC0 Public Domain Dedication. **d** Quantile–quantile plot of the social cost distribution against the population distribution. The three lines represent comparisons of social costs from CO$_2$-related climate change, PM$_{2.5}$-related health damage, and integrated costs. Source data are provided as a Source Data file.

generation significantly contributes to the social costs of CO$_2$-related climate change, and its health co-benefits are well studied[16,34]. However, in terms of health damage attributable to PM$_{2.5}$ exposure, its contribution is comparatively less than that of industrial and domestic bituminous coal combustion and diesel vehicle emissions. Our assessment indicates that phasing out existing coal-fired power plants in mainland China in 2017 would reduce national losses from health damage by only 5% due to their minor role in reducing air pollutant emissions. Moreover, the health co-benefits of decarbonizing large-scale boilers could diminish if coal-fired power plants extensively adopt ultralow emission control technologies[35]. Scenario analyses based on Nationally Determined Contributions (NDCs) also reveal that climate policies alone are insufficient to achieve the air quality standards necessary for public health protection[11,14]. Conversely, despite a minimal contribution to CO$_2$ emissions, fuel switches in the domestic sector are prioritized when health damage is integrated into the social cost assessment. The social cost of health damage significantly contributes to the integrated costs in regions such as Beijing, Hebei, and Shanxi. The clean winter heating plan in Northern China launched in 2017, which targets coal-fired household stoves in Beijing, Tianjin, and 26 surrounding cities, has reduced domestic coal combustion in these

areas[36]. Domestic solid fuel combustion also contributes more than 20% of the total integrated costs in northern provinces such as Jilin, Heilongjiang, and Gansu, justifying expanded residential fuel switch programs in less-developed northern regions.

A comparison of integrated costs at 36 km by 36 km resolution facilitates the development of nuanced decarbonization strategies tailored to regional, provincial, and city levels. The spatial control priorities based on integrated costs differ from those guided solely by monetized climate impacts. Unlike long-term greenhouse gases, PM$_{2.5}$ and its precursors, which have shorter lifetimes, significantly impact health depending on the proximity of the source[37]. Including monetized health damage in integrated costs highlights the need to focus on emission control in densely populated areas such as Shanghai, Chongqing, Zhengzhou, and the BHT city cluster. Urbanization offers opportunities for climate change mitigation through improved energy efficiency and public transportation[38] but also concentrates emissions, increasing air pollution-related health risks in cities[39]. A lower dependency of health damage on population density in cities with minimal domestic solid fuel consumption endorses initiatives such as the clean heating campaign to mitigate urbanization's impact on air quality. These findings support the integration of the Low-Carbon Cities

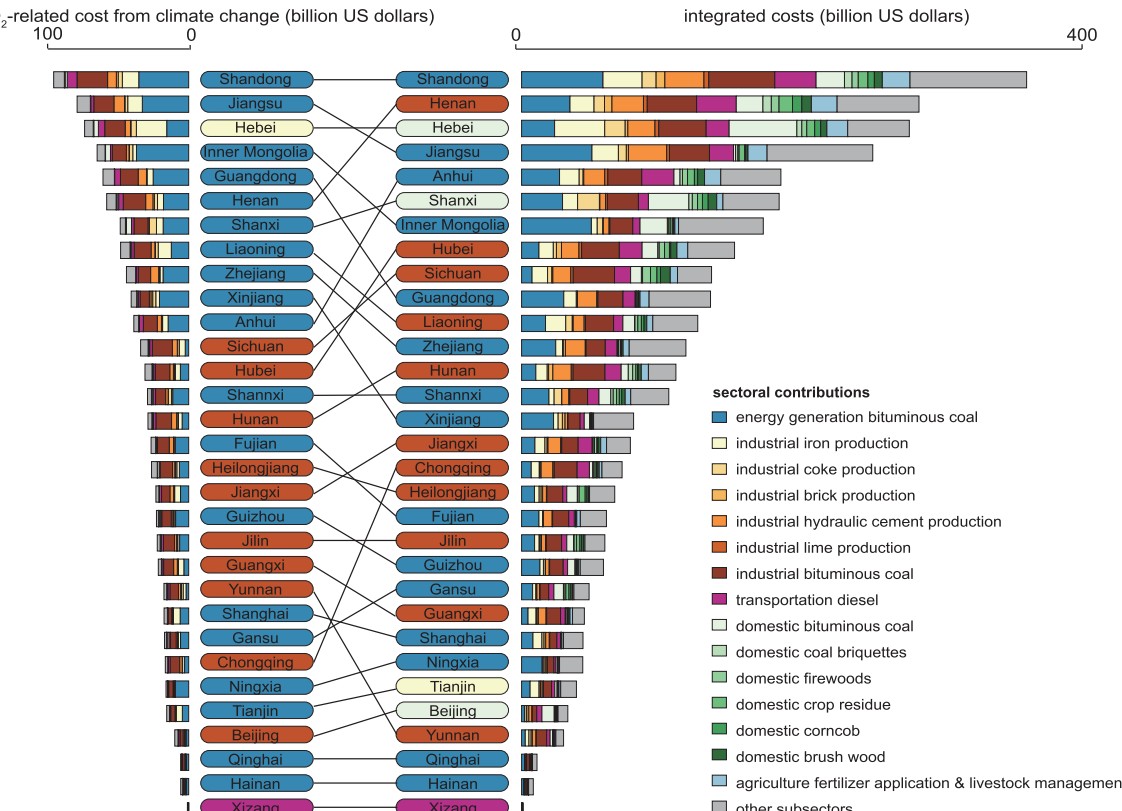

**Fig. 4 | Sectoral contributions to social cost from CO₂-related climate change and integrated cost in each province in mainland China.** The provinces were ranked in descending order based on their contributions to the social cost of CO₂-related climate change (left column) and the integrated cost (right column). The sectoral contributions in each province are plotted in the bar chart. The provinces are color coded based on their predominant subsectors. Source data are provided as a Source Data file.

Program into China's NDC and its global application[40,41], emphasizing population density in pilot program selection for enhanced health co-benefits. Neglecting health impacts in mitigation investments, such as carbon trading, could lead to prioritizing CO₂ emission reductions in less-developed western regions because reducing emissions is more cost-effective in regions with outdated combustion and control technologies[42].

The sectoral and spatial disparities in the source contributions to health damage and CO₂ emissions identified in our study underscore the necessity for integrated assessments at a high spatial resolution to inform balanced strategies that address both air pollution and climate change mitigation. CMAQ Adjoint's ability to assess the marginal benefits of emission reductions on a granular scale enables high-resolution source attribution, accounting for emission profiles, population density, and atmospheric conditions. The Sichuan Basin, for instance, is highlighted as an important contributor to health damage but not to CO₂ emissions, illustrating the nuanced insights provided by adjoint-based source attribution, which accounts for the complex interplay of factors, including population density and terrain, which can exacerbate PM₂.₅ accumulation and secondary aerosol formation[43].

Our results are subject to several limitations and uncertainties. First, limited by the availability of up-to-date high-resolution emission inventories with detailed source information, our source attribution analysis is based on adjoint sensitivities simulated for the year 2017. Considering China's recent advancements in air pollution control, such as the adoption of ultra-low emission standards in power plants and the clean heating campaign, changes in the emission profile could alter the source attribution results. Second, our health impact assessment focuses exclusively on premature deaths due to ambient PM₂.₅ exposure, and excludes indoor exposures. The contribution of emissions from domestic solid fuel combustion to health damage is likely to be more pronounced when considering indoor exposure, given that both emission and exposure predominantly occur within indoor environments[44–46]. Third, the integrated cost assessment is sensitive to the monetized values of health damage and climate impacts. There is considerable variability in both the VSL and the SCC estimates, spanning several orders of magnitude. The global SCC used in this study is greater than the country-level SCC, and reflects the worldwide impact of CO₂ emissions[47]. This implies that the relative importance of health damage, and thus the focus on mitigating pollution from heavily polluted sources, would be more pronounced when considering local social costs. Future studies constraining the monetized estimates of VSL and SCC are fundamental to ensuring robust and informed policymaking.

## Methods
### Model configuration
We used CMAQ v.5.0 with its recently developed adjoint model (CMAQ Adjoint) to simulate ambient PM₂.₅ concentrations and the sensitivity of population health loss to related air pollutant emissions in China in 2017. The CB05-AERO5 chemical mechanism was used in the CMAQ simulations. The CMAQ Adjoint implements the full adjoint of CMAQ v.5.0, including the discrete adjoint for gaseous chemistry, aerosol formation, cloud chemistry and dynamics, and diffusion and the continuous adjoint for advection[48]. Its detailed treatment of multiphase reactions for size-resolved aerosols enables adjoint sensitivity analysis of PM₂.₅-related health impacts. CMAQ has been widely applied and validated for simulating ambient PM₂.₅ concentrations and

assessing source impacts across various regions, including China[49,50]. The CMAQ Adjoint has also been successfully applied in source attribution for air pollution-related health impacts and inverse modeling[51,52].

The meteorological fields used to drive the CMAQ and CMAQ Adjoint simulations were downscaled from 0.5° × 0.5° global weather forecast products from the National Centers for Environmental Prediction Global Forecast System using the Weather Research Forecasting Model v.3.4.1[53,54]. The 2017 AiMa emission inventory was used as the emission input[55]. The inventory provides constrained bottom-up emission data using ground measurements and satellite observations. It has been widely used in previous studies and air quality forecasting services in China[53,56]. The study domain covers China at a 36 km by 36 km horizontal resolution (Supplementary Fig. 8), and 13 vertical layers that extend ~16 km above the ground.

We evaluate the performance of the CMAQ model by comparing the simulated and observed $PM_{2.5}$ concentrations at the 1504 monitoring sites across China. The comparison showed satisfying model performance according to the recommended benchmarks, with a Pearson correlation coefficient (r) of 0.76 (Supplementary Fig. 9)[57]. Furthermore, the adjoint model performed well when validated against forward sensitivities calculated using the finite difference and complex variable methods[48].

## CMAQ Adjoint analysis for health impacts
The adjoint model calculates the sensitivity of all sources by propagating the associated forcing to the receptors backward through time and space for a defined scalar cost function of the concentration field. Based on the adjoint equations derived from the atmospheric diffusion equations, the forcing term can be computed as the gradient of the cost function with respect to concentrations[48,58]. In this study, we defined a cost function, $J$, which represents the nationwide premature death attributable to long-term ambient $PM_{2.5}$ exposure in China in 2017 as shown in Eqs. (1)–(3), following the integrated exposure-response (IER) models in global burden of disease[7].

$$J = \sum_{i,g} B_i \cdot pop_g \cdot \left[1 - RR_i(z_g)^{-1}\right] \tag{1}$$

$$RR_i(z_g)^{-1} = 1 + \alpha_i \cdot \left(1 - e^{-\beta_i \cdot z_g^{\gamma_i}}\right) \tag{2}$$

$$z_g = \max(z_g - z_{cf}, 0) \tag{3}$$

where $B_i$ is the regional background mortality rate for each of the five disease-burden causes, including ischemic heart disease, cerebrovascular disease (stroke), chronic obstructive pulmonary disease, and lung cancer for adults older than 25 years and acute lower respiratory infections for children under five[59]; $pop_g$ is the gridded population data in mainland China[60]; and $RR_i$ is the IER function for evaluating the relative risk of disease i, as defined by the parameters $\alpha_i$, $\beta_i$, and $\gamma_i$, and the counterfactual $PM_{2.5}$ concentration, $z_{cf}$[7]. The background mortality rates and IER model parameters for each cause are listed in Supplementary Table 3. The parameter $z_g$ is the annual mean $PM_{2.5}$ concentration in grid $g$ obtained from the CMAQ 1-year forward simulation.

## Highly resolved emission inventory
The AiMa emission inventory categorizes emissions into coarse sectors, including electric, industrial, domestic, transportation, agricultural, solvent usage, fugitive dust, and biomass burning. To quantify the emissions of air pollutants and $CO_2$ in the detailed subsectors, we coupled anthropogenic emissions in the AiMa

inventory with the detailed bottom-up emission inventories generated by the Global Emission Modeling System (GEMS). The GEMS inventories feature detailed sector and fuel information and local and updated emission factors[61,62]. The emissions of each sector were apportioned into subcategories based on the relative contribution of each subcategory derived from the GEMS inventory. Energy generation, industrial, domestic, transportation, and agricultural activities were divided into 6, 24, 15, 5, and 3 subcategories, respectively. The detailed subcategories for each sector are listed in Supplementary Table 1.

## Source attribution analysis for health damage
We quantify the health damage attributable to major anthropogenic sectors and their subsectors at a monthly resolution by combining the sectoral air pollution emission amount and the adjoint sensitivity as Eq. (4):

$$M_{g,m,s} = \sum_p \frac{\partial J}{\partial E_{p,g,m}} \cdot \Delta E_{p,g,m,s} \tag{4}$$

where $\frac{\partial J}{\partial E_{p,g,m}}$ represents the marginal change in the annual premature death attributable to ambient $PM_{2.5}$ exposure in China in 2017 per unit change in emissions of air pollutant p in grid g and month m. The total health damage from sector s equals the sum of premature deaths attributable to the emissions of each primary $PM_{2.5}$ species and its precursors, including OC, EC, other primary $PM_{2.5}$, $SO_2$, $NO_x$, and $NH_3$. The contributions from volatile organic compounds emissions were also not discussed because of their minor contributions to $PM_{2.5}$-related health damage (Supplementary Fig. 10). The monthly averaged adjoint sensitivities for each air pollutant species ($\frac{\partial J}{\partial E_{p,g,m}}$) were calculated as emission-weighted averages from the three-dimensional daily adjoint sensitivity outputs from the CMAQ Adjoint.

## Monetized assessment and integrated costs
The social cost of $PM_{2.5}$ exposure-related health damage can be evaluated as the monetized value of premature deaths based on the economic VSL. The VSL in China in 2017 ($VSL_{c,2017}$) was calculated using the benefit-transfer approach as Eq. (5):

$$VSL_{c,2017} = VSL_{OECD} \cdot \left(\frac{Y_{c,2017}}{Y_{OECD}}\right)^e \tag{5}$$

where $VSL_{OECD}$ is the baseline VSL estimated for OECD countries, which equals US\$3.83 million; $Y_{OECD}$ is the average GDP per capita for the corresponding OECD countries in 2011[63]; $Y_{c,2017}$ is the GDP per capita of China in 2017 (adjusted to 2011 US dollars at purchasing power rates); and $e$ represents the income elasticity of the VSL and is assumed to be 1.2, as suggested by the World Bank[63]. For ethical reasons, the uniform $VSL_{c,2017}$ was applied to all grids, and monetized health impacts were assessed by multiplying the premature deaths with the VSL. The VSL values derived from the OECD benchmarks represent the lower end of the VSL estimates. The corresponding social cost of $CO_2$-related climate change for each subsector can be evaluated based on the $CO_2$ emission amount and SCC. Global values of the SCC for 2020 were employed to measure the direct social benefits, which account for the climate change damage avoided worldwide for a ton of $CO_2$ emission reduction in 2020. The latest SCC estimate at a 2.5% discount rate from the USEPA was adopted and adjusted to the 2011 value, which equals 100 US dollars per ton of $CO_2$ emission[64]. The SCC estimate is close to a recent expert elicitation estimate (approximately 80–100 US dollars when outliers were trimmed)[65]. We further evaluate the integrated costs as the sum of the social costs of health damage and climate change.

The impacts of VSL and SCC uncertainties on the monetized assessment are evaluated by perturbing the adopted values. A high estimate of VSL was calculated using the baseline VSL suggested by the USEPA, the average GDP per capita for the U.S., and an income elasticity of 0.5[66]. The VSL was calculated to be 4.36 million US dollars in 2011. Another VSL estimate based on a contingent valuation study in six representative cities in China (0.66 million in 2011 US dollars) is viewed as a low estimate of the VSL[67]. The SCC is perturbed from 10 to 1000 US dollars per ton of $CO_2$ emission[64] for uncertainty assessment.

## Reporting summary

Further information on research design is available in the Nature Portfolio Reporting Summary linked to this article.

## Data availability

Input datasets related to this paper are publicly available. Demographic data used in this study can be accessed via https://landscan.ornl.gov/. Other data supporting health damage assessment and monetized social cost assessments are available within the article and Supplementary Information. China Multi-Regional Input–Output Table is available at http://www.ceads.net/data/input_output_tables/. Emission inventories are available at https://gems.sustech.edu.cn/home. The data that support the plots within this paper are provided in the Source Data file. Source data for the ratio of gridded contributions to $PM_{2.5}$ exposure-related health damage and $CO_2$ emissions, alongside the gridded contributions to social costs from health damage, $CO_2$-related climate change, and integrated costs can be accessed via Zenodo: https://doi.org/10.5281/zenodo.11632297. Source data are provided with this paper.

## Code availability

The CMAQ Adjoint 5.0 model code can be accessed at https://github.com/USEPA/CMAQ_ADJOINT (https://doi.org/10.5281/zenodo.3780216). MATLAB R2021a was used for source attribution analysis in this study. The source codes utilized in this study can be assessed on https://doi.org/10.5281/zenodo.11632297.

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

## Acknowledgements

Y.L.C. acknowledge funding from the National Natural Science Foundation of China (42207116), Y.L.C., H.S and G.S. acknowledge funding from the Ministry of Science and Technology of the People's Republic of China (2023YFE0112900), H.S. acknowledge funding from the National Natural Science Foundation of China (42192510), Y.L.C. acknowledge funding from the fellowship of China Postdoctoral Science Foundation (2021M701573), H.S. acknowledge funding from the Shenzhen Key Laboratory of Precision Measurement and Early Warning Technology for Urban Environmental Health Risks (ZDSYS20220606100604008), Shenzhen Science and Technology Program (JCYJ20220818100611024), Department of Science and Technology of Guangdong Province (2021B1212050024 and 2020B1111360001), Department of Education of Guangdong Province (2021KCXTD004), High level of special funds (G03050K001 and G030290001), and Center for Computational Science and Engineering at Southern University of Science and Technology.

## Author contributions

Y.L.C. conceived and initiated the study. Y.L.C. and H.S. processed and analyzed the data. H.S., J.M., and S.T. assisted in the development of the model framework. Y.L.C. drafted the paper, and H.S., G.S., and S.T. participated in the result discussions. Y.F.C., A.G.R., S.Z., and A.H. provided critical revisions.

## Competing interests

The authors declare no competing interests.
