## [Peer Review File · Nature Communications]

REVIEWER COMMENTS

Reviewer #1 (Remarks to the Author):

Overall, the research is novel, relevant to scientists and policymakers, and has robust methodology. However, there were major issues with verbiage and sentence structure that made the manuscript challenging to read. Before the paper is published, it would benefit from professional writing assistance. Below, I have highlighted additional ways to improve the manuscript.

Air pollutants contribute to both premature mortality and climate change. If meaning health harming air pollutants (i.e. criteria air pollutants, as defined by the U.S. Clean Air Act), you need to specify.

Similarly, you use the phrase “direct social benefits” to mean the monetized climate impacts from applying the SCC to the reduced carbon emissions. However, there are social benefits from reduced PM2.5 pollution too. You need to specify climate change social benefits, or define the term.

MHB (monetized health benefits) is confusing because HB is referring to marginal health benefits. I would suggest not using these abbreviations at all.

The analysis only considers health co-benefits per ton of CO₂ (or monetized health co-benefits per ton of CO₂). These metrics do not get at the magnitude of decarbonization impacts on health. For example, gasoline combustion may have a lower health impact per ton of CO₂ compared to aviation transportation, but likely there are more cars that lead to a bigger overall health and climate impact from complete decarbonization. Your analysis should be re-worked to include data and analysis that captures the magnitude component.

Introduction

Give more background on the air pollutants that cause premature mortality and whose reduced emissions would yield health benefits. Which pollutants are you modeling and why. Cite papers that connect those pollutants to health impacts. Write out chemicals before using the abbreviation (OC, EC, PM2.5, SO₂, and NO_x). Mention that SO₂ and NO_x react to form secondary PM2.5.

Specify that the health co-benefits analysis only considering the health impacts from ambient PM2.5 exposure (I was surprised given the air quality model used).

Results

The first paragraph (lines 77 – 93) reads like methods. If the methods section will come last, perhaps you could move some of this to the last paragraph in the Introduction as a roadmap of where the paper is going.

Line 113 – Carbon neutral is in quotes – why? Add further explanation and citation.

Line 113 – Specify/define the non-synergistic subsectors.

You define domestic sector as the combined residential and commercial sectors, however only the residential sector is in the source category in Figure 1a. This was confusing to me – the text and the figure labels should match.

Also, specify what type of activities are included in the residential sector. Does it include electricity generation for buildings? Or coal combustion in homes for cooking and/or heating? What do you mean by poor burning conditions in line 97?

Figure 1 is well made. You pack a lot of information in it!

It should be labeled a-c, with a sublabel for each of the pie charts. The top pie chart – is that the inventory used in the modeling? Add a description of that chart in the caption and talk about it in the text.

The analysis of population density and domestic coal consumption is really interesting.

Figure 2 – having a consistent x-axis across the plots in 2b could be useful for comparison.

Line 172 – rename this section something about evaluating/comparing monetized benefits from reduced harmful air pollutants and avoided CO2 emissions

Lines 222-224 – This sentence makes it seem like that did happen in 2017 as opposed to what was found in the modeling.

Line 217 – start a new section and use the header from line 172. Breaking up these sections will clarify that a new analysis is starting.

Line 217 – First sentence should be clearer about what you mean by integrated benefits. Are you summing the two? Add a reference to Figure 3a.

Line 267 – Great note about the importance of the atmospheric conditions being an important consideration in this analysis.

Figure 3. Remove the “_” from the labels.

The lines connecting the cities could be thicker and I can't see the difference between black and blue. The lines could all be the same color, it doesn't add much depth to the figure analysis for them to be different.

Overall, this section reads like a Results and Discussion section. If this is intended to just be Results, a good portion of the information should be moved to another section.

Discussion and Conclusion

Lines 294 – 295 How do your findings inform localized decarbonization plans? At what jurisdictional level is “local”?

Line 309 – what do you mean by addressed?

Line 312 - Add results and citations to studies that consider the health impact of indoor domestic pollution in China.

Much of what is included in the Results should be moved to the Discussion. That will help flesh out this section.

Other components to consider: who is the integrated analysis intended for? Policymakers, city officials, etc.

There should justification for this use of the CMAQ model as well as discussion of the model and integrated benefits methods limitations. Not including indoor air pollution exposure would be a part of that.

Reviewer #2 (Remarks to the Author):

Thank you for the opportunity to review this manuscript that estimates the air quality-related health co-benefits of reducing CO₂ emissions in mainland China across 35 subsectors, using CMAQ Adjoint and a high-resolution emissions inventory disaggregated into detailed sectors and fuels.

The modelling appears sound, and the authors are correct that targeting specific sectors and locations for emissions reduction is a more efficient way of improving public health. However, I believe that the results as they stand are very difficult to interpret, in large part because they are expressed in terms of "HB", which represents the air quality-driven health burden of each source divided by the CO₂ emissions from that source.

The authors seem to be suggesting that, so long as we are agnostic as to which ton of CO₂ emissions to abate, we might as well do this from a source and location where there are higher health benefits. If this was the right analytical frame, then HB would be a meaningful unit to report. This frame is unwarranted for several reasons: (1) we are not agnostic as to which ton of CO₂ emissions to abate; some are far more costly to abate than others (both in terms of implementing the abatement technology and the potential aggregate effects on economic output); (2) even if we were agnostic as to which ton of CO₂ emissions to abate, it is not clear that the most straightforward way of doing so would reduce the air pollution health burden by the same amount. For example, CCS on coal plants will reduce CO₂ but leave air quality unchanged; SO₂ scrubbers will improve air quality but leave CO₂ unchanged. (There are many more such examples across the subsectors listed.)

The claim that for every % reduction in CO₂ emissions there is an equivalent % reduction in air pollutant emissions across all subsectors and locations is really only warranted in the very particular case of 'phasing out' that pollution source (e.g., avoiding burning fuel). This seems to be the main concern of the paper, leaving out all the other policy options, some of which may be more likely and cost-effective in the short term (and which will, if implemented, completely change the results of this analysis).

The analysis also leaves out "non-synergistic" subsectors---even though these are responsible for over a quarter of the deaths---even though (e.g., in the case of agriculture) these subsectors can be large sources of non-combustion-related, non-CO2-related greenhouse gas emissions. In all, this paper has a somewhat narrow conception of air quality and climate mitigation options (only being about phasing out fuel combustion), and a very simplistic view of how those would be implemented in policy (no consideration of economic costs/benefits, and only concerned with the sources to target rather than the actual mitigation options).

To overcome these issues, I recommend that the authors (a) directly report the health damages alongside the CO2 emissions, rather than the HB metric; (b) expand their analysis to include "non-synergistic sectors" and more greenhouse gas emissions (not just CO2), or make it clear from the outset that they are only considering more traditional sources of air pollution (e.g., fuel combustion in industry/transport); (c) add some more context related to the economic costs and benefits, if not of mitigation options then at least of sectoral output (e.g., how much does each sector currently contribute to GDP?).

Some more detailed comments:

"35 emission sources"---do you mean subsectors? "Sources" is vague.

"For example, 45 following the RCP4.5 scenario (which depicts a moderately warming future), researchers have 46 estimated that CO2 emission reductions can help prevent 1.3 million premature deaths globally 47 in 2050 by reducing air pollutant emissions simultaneously." What is the citation for this statement? Is this source 10 (Shindell et al., 2018)?

- Figs 1 and 3 are very difficult to interpret. There is a lot going on in each of them. I suggest having figures that each present one finding very clearly and convincingly.

Nature Communications manuscript NCOMMS-23-32437-T

Title: Substantial differences in source contributions to carbon emissions and health damage necessitate balanced synergistic control plans in China

REVIEWER COMMENTS

Reviewer #1 (Remarks to the Author):

Comment 1

Overall, the research is novel, relevant to scientists and policymakers, and has robust methodology. However, there were major issues with verbiage and sentence structure that made the manuscript challenging to read. Before the paper is published, it would benefit from professional writing assistance. Below, I have highlighted additional ways to improve the manuscript.

[Response]

Thank you for your thorough review and overall support of our study. We appreciate your criticism regarding the manuscript's readability. In response to your concerns, we have engaged the AJE language editing service to refine the manuscript's language and structure. The editing certificate has been attached. We have also meticulously addressed the additional comments you provided, as detailed in the point-by-point responses below.

Comment 2

Air pollutants contribute to both premature mortality and climate change. If meaning health harming air pollutants (i.e. criteria air pollutants, as defined by the U.S. Clean Air Act), you need to specify.

[Response]

We apologize for the confusion. There was an editing problem in the sentence in line 96 where we mention "air pollutants". The sentence has been revised as "The source contribution of health-threatening air pollutants differs from that of CO₂, although both mainly originate from fossil fuel combustion."

We also define the air pollutants included in the health impact assessment in the last paragraph of the Introduction as follows.

"The analysis includes impact of emission from speciated primary PM_{2.5}, which include organic carbon (OC), elementary carbon (EC), and other primary PM_{2.5} particles, as well as from principal precursors of secondary PM_{2.5}, which include sulfur dioxide (SO₂), nitrogen oxides (NO_x), and ammonia (NH₃)."

Comment 3

Similarly, you use the phrase "direct social benefits" to mean the monetized climate impacts from applying the SCC to the reduced carbon emissions. However, there are

social benefits from reduced PM_{2.5} pollution too. You need to specify climate change social benefits, or define the term.

[Response]

Thank you for the comment. The revised manuscript focused on source contributions to CO₂ emissions and health damage instead of the health co-benefit (HB) metric as suggested by the second reviewer. In response to this comment, we replaced the phrase “direct social benefits” to “social cost from CO₂-related climate change” and the phrase “health co-benefits” to “social cost from PM_{2.5} exposure-related health damage” throughout the text for clarification.

Comment 4

MHB (monetized health benefits) is confusing because HB is referring to marginal health benefits. I would suggest not using these abbreviations at all.

[Response]

Thank you for providing this suggestion. As suggested by comment 5 and relevant comments from Reviewer 2, we no longer report the metric of HB (or MHB) as health co-benefits per ton of CO₂ (or monetized health co-benefits per ton of CO₂) in the revised text. Instead, we directly report the source contributions to PM_{2.5} exposure-related health damage and CO₂ emissions. Therefore, we replaced the abbreviations and used the phrase “health damage”, “monetized health damage”, “social cost from PM_{2.5} exposure-related health damage”, or “source contribution to health damage” depending on the context of the revised manuscript.

Comment 5

The analysis only considers health co-benefits per ton of CO₂ (or monetized health co-benefits per ton of CO₂). These metrics do not get at the magnitude of decarbonization impacts on health. For example, gasoline combustion may have a lower health impact per ton of CO₂ compared to aviation transportation, but likely there are more cars that lead to a bigger overall health and climate impact from complete decarbonization. Your analysis should be re-worked to include data and analysis that captures the magnitude component.

[Response]

We agree with the reviewer that the magnitude of health and climate impacts are important in analyzing the source contributions. To address this, we have re-performed the health impacts analysis to include both synergistic and non-synergistic subsectors, and report the magnitude of impacts in the revised figures and text.

Subsequently, Figure 1 has been updated to enable a direct comparison of the impacts on premature deaths and CO₂ emissions across subsectors, as illustrated in panel c. The corresponding narrative, specifically lines 118-131, has been meticulously revised to articulate the considerable variations in source contributions to health damage and CO₂ emissions. This section now reads:

“Seven subsectors were identified as significant contributors to health damage, each of which was attributable for more than 100,000 premature deaths (**Figure 1c**). These subsectors include bituminous coal combustion in the domestic, energy generation, and industrial sectors; emissions from diesel-powered vehicles; and activities in hydraulic cement production, iron and steel production, as well as agricultural fertilizer application and livestock management. While these subsectors comparably contribute to health damage, their contributions to CO₂ emissions differ substantially. For instance, agricultural activities significantly contribute to PM_{2.5}-related health damage from NH₃ emissions without being major contributors to CO₂ emissions directly. This discrepancy is also observed in the analysis of CO₂ emissions from other subsectors. For example, the CO₂ emissions from domestic bituminous coal combustion are 52-93% lower than those from the remaining five subsectors. In terms of the speciated emission contribution to health damage, more than half of the health damage from hydraulic cement and iron production, as well as domestic bituminous coal combustion, is attributed to primary PM_{2.5} emissions. In contrast, health damage from bituminous coal combustion in the energy generation and industrial sectors, as well as from diesel vehicles, is predominantly attributable to SO₂ and NO_x emissions (**Figure S2**). The impact of various fuel types and industrial processes on premature deaths within a sector also shows substantial disparities. For example, diesel-powered vehicles, which are responsible for 210,000 premature deaths, contribute eightfold more to health damage than do gasoline-powered vehicles, which emphasizes the great importance of diesel vehicle electrification in reducing air pollution and associated health risks. Overall, the ratio of source contributions to health damage versus to CO₂ emissions varies extensively across subsectors, ranging from 0.094 in the industrial combustion of dry natural gas to 22 in the domestic combustion of unorganized waste. This noteworthy variation in source profiles for health damage and CO₂ emissions underscores that strategies aimed at reducing air pollution-related health damage may not always align with those targeting decarbonization. This necessitates the formulation of coordinated, synergistic control plans to effectively balance the two objectives.”

Furthermore, Figure 2b has been revised to depict the magnitude of provincial contributions to health damage and CO₂ emissions across five sectors, with the related sentences in lines 142-149 refined to directly contrast the magnitudes of these contributions. This section now states:

“Spatial heterogeneity in the source contributions to health damage and CO₂ emissions is also evident within individual sectors. For instance, in the energy generation sector, Inner Mongolia contributes marginally more to CO₂ emissions (3.8%) than Shandong (3.5%), yet its impact on health damage (0.75%) is significantly lower than Shandong's (1.3%) (**Figure 2b**). Similarly, in the industrial sector, Guangdong's CO₂ emissions surpass those of Anhui and Hebei (18%), yet its health damage contribution is merely half of that observed in the latter provinces. This spatial variability is influenced not only by population density but also by the intensity of air pollutant emissions (**Figure S4**). For example, Sichuan, with its lower coal quality leading to higher emissions of PM_{2.5} and SO₂ per ton of CO₂ emission, exhibits the most pronounced disparity in the

energy generation sector. Similarly, the elevated PM_{2.5} emissions associated with clinker production in the hydraulic cement manufacturing make Anhui and Hebei the regions with the highest discrepancy in the industrial sector.”

Figures 1 and 2 have been revised as follows,

Figure 1. Disparities in source attributions of CO₂ emissions and health damage. (a) Source contributions to CO₂ emissions in China in 2017 based on the coordinated CO₂ emission inventory with the air pollutants. The lower bar represents the contributions of different sectors, and the upper bar represents the contributions of different fuel/process categories. **(b)** Source contributions to premature deaths attributable to long-term ambient PM_{2.5} exposure in China in 2017 based on the CMAQ adjoint simulation. The left bar represents the contributions of different sectors, and the right bar represents the contributions of different fuel/process categories. The shaded areas are non-synergistic subsectors without direct CO₂ emissions. **(c)** Comparison of impacts on health damage and CO₂ emissions for each subsector. The corresponding monetized impacts are also shown on the minor axis based on a uniform statistical life value (1.33 million USD per premature death) and social cost of carbon (100 USD per ton of CO₂ emission). The abbreviations for the fuel/process categories are listed in **Table S2**.

Figure 2. Disparities in spatial distributions of CO₂ emissions and health damage. (a) Ratio between the gridded contribution to PM_{2.5} exposure-related health damage and to CO₂ emissions. The provincial boundary shapefile is obtained from Harvard Dataverse (<https://doi.org/10.7910/DVN/DBJ3BX>) and is publicly available under the Creative Commons CC0 Public Domain Dedication. **(b)** Comparison of provincial contributions to health damage and CO₂ emissions in each sector. The red dots represent the ratio of provincial contributions to health damage versus to CO₂ emissions. The provinces are ranked from large to small in population size. The abbreviations of the provinces are listed in **Table S3**. **(c)** Correlation between population density and the ratio of each city's contributions to health damage versus to CO₂ emissions. The dot color represents the domestic solid fuel usage fraction, which is defined as the fraction of energy consumed as solid fuels in the domestic sector out of the total province-level energy consumption. The solid line and dashed line represent the correlations for cities with domestic solid fuel consumption above and less than 1%, respectively, of the total provincial energy consumption.

Comment 6

Introduction

Give more background on the air pollutants that cause premature mortality and whose reduced emissions would yield health benefits. Which pollutants are you modeling and why. Cite papers that connect those pollutants to health impacts. Write out chemicals before using the abbreviation (OC, EC, PM_{2.5}, SO₂, and NO_x). Mention that SO₂ and NO_x react to form secondary PM_{2.5}.

[Response]

In response to the reviewer's request for additional background on the air pollutants linked to premature mortality and their health benefits upon emission reduction, we have updated our manuscript to provide a more comprehensive overview. Specifically, line 39 has been revised to highlight the critical role of PM_{2.5} in air pollution-related premature mortality in China, detailing the associated health risks as follows:

“Notably, around 80% of these deaths are linked to ambient particulate matter with a diameter of 2.5 micrometers or smaller (PM_{2.5}), which exacerbates mortality risks associated with ischemic heart disease, cerebrovascular disease, chronic obstructive pulmonary disease, lung cancer, and lower respiratory infections^{6,7}.”

Our analysis focuses on the sectoral contributions to premature mortality associated with ambient PM_{2.5} exposure, primarily due to its significant contribution to air pollution-related premature deaths. Therefore, we only include the premature deaths attributable to ambient PM_{2.5} exposure in the cost function defined for CMAQ Adjoint simulations.

In the revised text, we clarify that our health co-benefits analysis is confined to premature deaths attributable to ambient PM_{2.5} exposure. We detail the primary PM species and precursors assessed in quantifying the sectoral contributions. The chemicals are spelled out at their first appearance in the revised text (lines 80 - 82). NH₃ is included because we expand our discussion to include the non-synergistic subsectors, which includes agriculture NH₃ emissions. The sentence stating that contribution from NH₃ emissions is not included because the agriculture practices are not directly associated with CO₂ emissions in lines 414-416, and the associated Figure S9 is deleted accordingly.

“The health impact assessment focuses on nationwide premature mortality attributable to ambient PM_{2.5} exposure. Our coordinated source attribution analysis integrates coupled emission inventories for both air pollutants and CO₂ with an advanced adjoint tool developed for a regional air quality model (CMAQ Adjoint). This tool allows for the efficient calculation of backward sensitivities, thereby quantifying the impact of high-resolution emission changes on nationwide premature mortality counts. The analysis includes impact of emission from speciated primary PM_{2.5}, which include organic carbon (OC), elementary carbon (EC), and other primary PM_{2.5} particles, as well as from principal precursors of secondary PM_{2.5}, which include sulfur dioxide (SO₂), nitrogen oxides (NO_x), and ammonia (NH₃).”

References

6. USEPA. Integrated Science Assessment (ISA) for Particulate Matter (Final Report, 2019). Report No. EPA/600/R-19/188, U.S. Environmental Protection Agency (USEPA): Washington, DC, 2019. (Assessed on 02/15/2023).
7. Cohen, A. J.; Brauer, M.; Burnett, R.; Anderson, H. R.; Frostad, J.; Estep, K.; Balakrishnan, K.; Brunekreef, B.; Dandona, L.; Dandona, R.; Feigin, V.; Freedman, G.; Hubbell, B.; Jobling, A.; Kan, H.; Knibbs, L.; Liu, Y.; Martin, R.; Morawska, L.; Pope, C. A.; Shin, H.; Straif, K.; Shaddick, G.; Thomas, M.; van Dingenen, R.; van Donkelaar, A.; Vos, T.; Murray, C. J. L.; Forouzanfar, M. H., Estimates and 25-year trends of the global burden of disease attributable to ambient air pollution: an analysis of data from the Global Burden of Diseases Study 2015. *Lancet* **2017**, *389*, (10082), 1907-1918.

Comment 7

Specify that the health co-benefits analysis only considering the health impacts from ambient PM_{2.5} exposure (I was surprised given the air quality model used).

[Response]

We have clarified in the manuscript that our evaluation is exclusively focused on the health impacts from exposure to ambient PM_{2.5}. This specification is now explicitly stated in the revised last paragraph of the Introduction: as follows:

“The health impact assessment focuses on nationwide premature mortality attributable to ambient PM_{2.5} exposure.”

Regarding the use of the adjoint model in our study, it is important to note that this model necessitates a predefined cost function to generate the input forcing data. For our research, this function corresponds to the premature deaths related to ambient PM_{2.5} exposure. The evaluation of health impacts from other air pollutant exposures would necessitate separate model simulations, which could be the interest of future studies.

Comment 8

Results

The first paragraph (lines 77 – 93) reads like methods. If the methods section will come last, perhaps you could move some of this to the last paragraph in the Introduction as a roadmap of where the paper is going.

[Response]

Thank you for your insightful suggestion regarding the structure of the manuscript. Based on your recommendation, we have reorganized the content to enhance clarity and flow. The initial paragraph of the results (lines 77-93), which previously outlined our methodological approach, has been adapted to serve as a roadmap in the last paragraph of the Introduction. The rewritten last paragraph in the Introduction is as follows:

“In this study, we perform a detailed quantitative analysis to delineate the contributions

of diverse sectors and their associated fuel/process combinations to CO₂ emissions and the health damage related to air pollution in China. This analysis encompasses critical sectors such as energy production, industry, transportation, domestic activities (both residential and commercial sources), and agriculture. The health impact assessment focuses on nationwide premature mortality attributable to ambient PM_{2.5} exposure. Our coordinated source attribution analysis integrates coupled emission inventories for both air pollutants and CO₂ with an advanced adjoint tool developed for a regional air quality model (CMAQ Adjoint). This tool allows for the efficient calculation of backward sensitivities, thereby quantifying the impact of high-resolution emission changes on nationwide premature mortality counts. The analysis includes impact of emission from speciated primary PM_{2.5}, which include organic carbon (OC), elementary carbon (EC), and other primary PM_{2.5} particles, as well as from principal precursors of secondary PM_{2.5}, which include sulfur dioxide (SO₂), nitrogen oxides (NO_x), and ammonia (NH₃). Moreover, we propose using the integrated costs – representing the aggregate of monetized social costs from health damage and climate change – as a comprehensive metric for prioritizing sectoral and spatial control targets.”

Comment 9

Line 113 – Carbon neutral is in quotes – why? Add further explanation and citation.

[Response]

Thank you for pointing out the need for clarification regarding the use of quotes around “carbon neutral” in line 113. The quotation marks were initially used to indicate that the assumption of carbon neutrality in crop residues is subject to debate. Recent findings suggest that the removal of crop residues can diminish soil organic carbon levels, potentially increasing CO₂ emissions, as evidenced by Liska et al. (2014). Additionally, carbon emissions are incurred during the harvest, transportation, and conversion processes of these residues, as highlighted in Xing et al. (2021). In our study, we chose not to account for the life-cycle emissions of crop residues to maintain consistency with the emission calculations for other fuel types. To elucidate this point, we have added explanatory sentences in line 113, which read as follows:

“While the combustion of crop residues does emit CO₂, this subsector is considered “carbon neutral” based on the assumption that the released CO₂ is later reabsorbed through subsequent biomass regrowth²³. As in our approach for other fuel types, life cycle emissions from biofuel processing and land use changes are not included^{24,25}.”

References

23. Ru, M.; Tao, S.; Smith, K.; Shen, G.; Shen, H.; Huang, Y.; Chen, H.; Chen, Y.; Chen, X.; Liu, J.; Li, B.; Wang, X.; He, C., Direct Energy Consumption Associated Emissions by Rural-to-Urban Migrants in Beijing. *Environ. Sci. Technol.* **2015**, *49*, 13708-13715, doi:10.1021/acs.est.5b03374.
24. Liska, A. J.; Yang, H.; Milner, M.; Goddard, S.; Blanco-Canqui, H.; Pelton, M. P., Fang, X. X., Zhu, H., Suyker, A. E., Biofuels from crop residue can reduce soil carbon and increase CO₂ emissions. *Nat. Clim. Chang.* **2014**, *4*, 398-401.

25. Xing, X.; Wang, R.; Bauer, N.; Ciais, P.; Cao, J.; Chen, J.; Tang, X.; Wang, L.; Yang, X.; Boucher, O.; Goll, D.; Peñuelas, J.; Janssens, I. A.; Balkanski, Y.; Clark, J.; Ma, J.; Pan, B.; Zhang, S.; Ye, X.; Wang, Y.; Li, Q.; Luo, G.; Shen, G.; Li, W.; Yang, Y.; Xu, S., Spatially explicit analysis identifies significant potential for bioenergy with carbon capture and storage in China. *Nat. Commun.* **2021**, *12*, 3159.

Comment 10

Line 113 – Specify/define the non-synergistic subsectors.

[Response]

In response to this suggestion, we have refined the sentence in line 113 for greater specificity and expanded Table S1 to encompass all non-synergistic subsectors. Furthermore, we have revised the contributions of these subsectors to reflect corrections made in the calculation of emissions and health impacts associated with the iron and steel production process. The revised sentence in the manuscript now reads:

“Certain subsectors contribute to health losses without being directly linked to CO₂ emissions from fuel combustion. These include non-combustion industrial processes, agricultural fertilizer application and livestock management, and crop residue burning in the domestic sector (**Table S1**). While the combustion of crop residues does emit CO₂, this subsector is considered “carbon neutral” based on the assumption that the released CO₂ is later reabsorbed through subsequent biomass regrowth²³. As in our approach for other fuel types, life cycle emissions from biofuel processing and land use changes are not included^{24,25}. We term these subsectors “non-synergistic subsectors”, which account for 23% of the total premature deaths attributable to the five sectors assessed.”

Table S1 The monetized health damage, CO₂ emissions, integrated benefits, and their associated ranks for all source sectors assessed in this study

sector	fuel/process category	monetized health damages (rank)	monitized climate impacts (rank)	integrated costs (rank)
industry	bituminous coal	2.2×10^{11} (1)	2.6×10^{11} (2)	4.8×10^{11} (2)
	anthracite	9.8×10^9 (22)	8.7×10^9 (13)	1.9×10^{10} (20)
	coking coal	4.9×10^9 (26)	5.3×10^9 (16)	1.0×10^{10} (26)
	gas/diesel	2.6×10^{10} (16)	8.9×10^9 (12)	3.5×10^{10} (15)
	dry natural gas	1.4×10^9 (37)	9.0×10^9 (11)	1.0×10^{10} (24)
	crude oil	3.4×10^9 (31)	9.2×10^9 (10)	1.3×10^{10} (22)
	residue fuel oil	1.1×10^9 (39)	2.6×10^9 (21)	3.6×10^9 (35)
	industrial waste	1.5×10^4 (53)	6.2×10^4 (36)	7.7×10^4 (53)
	primary Al production	1.9×10^9 (33)	2.0×10^9 (22)	3.9×10^9 (34)
	hydraulic cement production	1.8×10^{11} (5)	7.6×10^{10} (4)	2.5×10^{11} (4)
	iron and steel production	1.3×10^{11} (7)	1.2×10^{11} (3)	2.4×10^{11} (5)
	coke production	5.7×10^{10} (10)	3.6×10^7 (7)	9.4×10^{10} (8)
	brick production	2.7×10^{10} (15)	1.2×10^{10} (9)	3.9×10^{10} (13)
	petroleum catalytic cracking	1.6×10^9 (36)	0 (N/A)	1.6×10^9 (38)
	lime production	2.8×10^{10} (14)	0 (N/A)	2.8×10^{10} (18)
	glass production	5.3×10^9 (25)	0 (N/A)	5.3×10^9 (32)
	fertilizer production	1.3×10^{10} (20)	0 (N/A)	1.3×10^{10} (21)
	ferroalloy production	8.9×10^9 (23)	0 (N/A)	8.9×10^9 (29)
	lead production	8.7×10^8 (42)	0 (N/A)	8.7×10^8 (45)
	magnesium production	1.6×10^8 (48)	0 (N/A)	1.6×10^8 (49)
zinc production	1.1×10^9 (38)	0 (N/A)	1.1×10^9 (42)	
ammonia production	1.6×10^9 (35)	0 (N/A)	1.6×10^9 (37)	
nickel and copper production	4.2×10^9 (28)	0 (N/A)	4.2×10^9 (33)	
natural gas production	1.2×10^{10} (21)	0 (N/A)	1.2×10^{10} (23)	
energy generation	bituminous coal	1.8×10^{11} (4)	3.5×10^{11} (1)	5.3×10^{11} (1)
	coking coal	2.6×10^7 (51)	5.1×10^7 (33)	7.6×10^7 (52)
	gas/diesel	6.6×10^7 (50)	1.7×10^8 (31)	2.4×10^8 (48)
	dry natural gas	2.3×10^9 (32)	8.0×10^9 (14)	1.0×10^{10} (25)
	residue fuel oil	3.6×10^8 (46)	1.1×10^9 (26)	1.5×10^9 (39)
	solid biomass	2.1×10^8 (47)	6.5×10^8 (27)	8.6×10^8 (46)

Table S1 The monetized health damages, CO₂ emissions, integrated benefits, and their associated ranks for all source sectors assessed in this study (continued)

sector	fuel/process category	monetized health damages (rank)	monitized climate impacts (rank)	integrated costs (rank)
domestic	bituminous coal	2.1×10^{11} (2)	2.6×10^{10} (8)	2.3×10^{11} (6)
	honeycomb/coal briquettes	2.9×10^{10} (13)	3.7×10^9 (18)	3.2×10^{10} (16)
	indoor firewood burning	5.0×10^{10} (11)	3.0×10^9 (20)	5.3×10^{10} (12)
	non-organized waste burning	9.3×10^8 (41)	2.4×10^7 (34)	9.5×10^8 (43)
	patent fuel	8.8×10^9 (24)	1.2×10^9 (25)	1.0×10^{10} (27)
	gas/diesel	1.8×10^9 (34)	1.4×10^9 (23)	3.2×10^9 (36)
	dry natural gas	1.0×10^9 (40)	2.3×10^8 (30)	1.2×10^9 (41)
	liquid petroleum gas	3.7×10^9 (30)	6.1×10^9 (15)	9.8×10^9 (28)
	gas works gas	8.6×10^8 (43)	4.9×10^8 (28)	1.4×10^9 (40)
	coke oven gas	6.1×10^8 (44)	3.4×10^8 (29)	9.5×10^8 (44)
	biogas	1.4×10^7 (52)	8.3×10^7 (32)	9.6×10^7 (51)
	indoor crop residue burning	6.4×10^{10} (8)	0 (N/A)	6.4×10^{10} (9)
	charcoal	4.2×10^8 (45)	0 (N/A)	4.2×10^8 (47)
	indoor corncob burning	3.8×10^{10} (12)	0 (N/A)	3.8×10^{10} (14)
	indoor brush wood burning	6.2×10^{10} (9)	0 (N/A)	6.2×10^{10} (10)
	transportation	gasoline	1.9×10^{10} (19)	3.97×10^8
diesel		2.0×10^{11} (3)	5.36×10^8	2.23×10^3
biogas		4.4×10^9 (27)	1.26×10^7	4.46×10^1
aviation		4.1×10^9 (29)	3.24×10^7	6.52×10^1
shipping		1.1×10^8 (49)	1.23×10^5	9.14×10^{-1}
agriculture	gas/diesel	2.6×10^{10} (17)	4.9×10^9 (17)	3.0×10^{10} (17)
	agriculture waste burning	2.0×10^{10} (18)	0 (N/A)	2.0×10^{10} (19)
	fertilizer application and livestock management	1.4×10^{11} (6)	0 (N/A)	1.4×10^{11} (7)

Comment 11

You define domestic sector as the combined residential and commercial sectors, however only the residential sector is in the source category in Figure 1a. This was confusing to me – the text and the figure labels should match.

Also, specify what type of activities are included in the residential sector. Does it include electricity generation for buildings? Or coal combustion in homes for cooking

and/or heating? What do you mean by poor burning conditions in line 97?

[Response]

The label in Figure 1a has been corrected to “domestic sector”. Thank you for noting the inconsistency between the text and Figure 1a.

Regarding the activities included within the domestic sector, we provide a comprehensive list in Table S1. This sector primarily involves direct combustion of coal, oil, gas, biomass, and waste for heating and cooking purposes. To avoid double-counting, we explicitly exclude electricity generation for buildings from this category, as such emissions are accounted for within the energy generation sector.

The term “poor burning conditions” referred to in line 104 has been clarified to better describe the suboptimal combustion processes often encountered in household stoves. These conditions are characterized by restricted air supply and poor mixing, which can lead to incomplete combustion and higher emissions of pollutants. The revised sentence now reads:

“Coal and solid-biomass combustion in the domestic sector contributed 38% of the total PM_{2.5} emissions and 22% of the total SO₂ emissions. This substantial contribution is attributed to the restricted air supply, poor mixing, and insufficient emission control in household stoves, coupled with the use of low-quality fuels²⁰.”

Comment 12

Figure 1 is well made. You pack a lot of information in it!

It should be labeled a-c, with a sublabel for each of the pie charts. The top pie chart – is that the inventory used in the modeling? Add a description of that chart in the caption and talk about it in the text.

[Response]

We are grateful for your feedback. The initial top pie charts in Figure 1 show source contribution to CO₂ emissions and PM_{2.5} exposure-related health damage, which is derived from coordinate CO₂ and air pollutant emission inventories used in this study and adjoint modeling. To enhance clarity, we have transformed these pie charts into bar charts so that we can align them along the corresponding axis showing the source contribution to health damage and CO₂ emissions from each subsector in the updated panel c. Furthermore, we have updated the source contributions in panels a and b following a correction in the emission data for the iron and steel manufacturing process. Each bar chart, along with the scatter plot, has been labeled from a to c, with concise descriptions added to each panel for better understanding. Please refer to the revised figure in our response to comment 5.

Additionally, the revised manuscript text now explicitly mentions that the source contributions to health damage and CO₂ emissions are derived from CMAQ adjoint simulations and coordinated emission inventories as follows.

The sentence in line 97-100 is revised as “An analysis of the coordinated emission

inventories for CO₂ and air pollutants for the year 2017 reveals that 86% of the total anthropogenic CO₂ emissions were predominantly from the energy generation and industrial sectors. Notably, coal combustion was the major contributor, accounting for 97% and 50% of the total CO₂ emissions in these two sectors, respectively (**Figure 1a**).”

Furthermore, the content in lines 105-108 has been updated as follows: “Adjoint-based source attribution indicates that more than a quarter of premature deaths attributable to ambient PM_{2.5} exposure originate from the domestic sector, despite its mere 4% contribution to CO₂ emissions (**Figure 1b**). In contrast, while the energy generation sector is an important contributor to CO₂ emissions, it accounts for only 10% of premature deaths.”

Comment 13

The analysis of population density and domestic coal consumption is really interesting.

[Response]

Thank you for your encouraging remarks. To address concerns from the second reviewer regarding the previously used metric, which quantified health co-benefits as avoided premature deaths per ton of CO₂ emission reduction, we have refined our analyses. The revised analysis now employs a more intuitive dependent variable: the ratio of contributions to health damage versus to CO₂ emissions for each city. We have re-examined its relationship with population density.

Our findings reveal a positive correlation between this ratio and population density, indicating that cities with higher densities tend to have a disproportionately greater impact on health relative to their CO₂ emissions. Notably, our analysis also highlights the significant role of domestic solid fuel use over coal consumption in driving this correlation. The inclusion of all non-synergistic subsectors has unveiled the substantial contribution of domestic biomass burning to health damage. Accordingly, we have updated the scatter plot and regressions in Figure 2. Please refer to the revised Figure 2 in our response to comment 5.

The manuscript has been amended in lines 156-170 to incorporate the updated analysis as follows.

“The color-coded scatter plot in **Figure 2c** illustrates the impact of reliance on solid fuels for domestic energy needs on the correlation between the ratio of contributions to health damage versus that to CO₂ emissions and population density. Notably, this correlation intensifies in cities with a greater dependence on solid fuels for domestic energy, explaining more than half of the variance in city-level ratios. Specifically, a 1% increase in population density correlates with a 0.47% increase in the ratio. Conversely, in cities where domestic solid fuel consumption constitutes less than 1% of total provincial energy usage, population density accounts for merely 4% of the ratio variance, with a 1% population increase leading to a modest 0.12% increase in the ratio. The higher ratio in Beijing than in other megacities, including Shanghai, Guangzhou, and Shenzhen, further demonstrates the critical role of domestic solid fuel consumption

(Figure S5). Although Beijing has already decommissioned its energy-intensive industries and coal-fired power plants, it is the only city among the four megacities whose contribution to health damage exceeds its contribution to CO₂ emissions. This increase in Beijing’s contribution to health damage was primarily attributed to the consumption of 1.8 Mt of raw coal by rural residents in 2017²⁸.”

The revised Figure S5 is as follows.

Figure S5 Comparison of city contribution to PM_{2.5}-related health damage and CO₂ emissions from top megacities in China. The stacked bars show source contributions to nationwide health damage (upper bars) and CO₂ emissions (lower bars) in each city. The red dots show the ratio between city contributions to health damage and CO₂ emissions.

Comment 14

Figure 2 – having a consistent x-axis across the plots in 2b could be useful for comparison.

[Response]

In alignment with the revisions discussed in our response to comment 13, where we transitioned away from employing the health co-benefit metric of avoided premature deaths per ton of CO₂ emission reduction, we have accordingly adapted Figure 2b. The updated figure now displays a comparison of the provincial contributions to PM_{2.5} exposure-related health damage and CO₂ emissions across different sectors. Following your suggestions, we have standardized the y-axis across all plots. However, we have opted for distinct scales on the minor y-axis for the ratio of source contributions to health damage versus to CO₂ emissions, due to the significant variability—spanning two orders of magnitude—across sectors. Please refer to the revised Figure 2 in our response to comment 5.

Comment 15

Line 172 – rename this section something about evaluating/comparing monetized benefits from reduced harmful air pollutants and avoided CO2 emissions

[Response]

In response to your suggestion, we have revised the title of the section spanning lines 174 to 216 to “Monetized social costs of CO₂ emissions and PM_{2.5} exposure-related health damage”.

Comment 16

Lines 222-224 – This sentence makes it seem like that did happen in 2017 as opposed to what was found in the modeling.

[Response]

In response to your comment, we have revised the sentence to more accurately convey that the integrated benefits were outcomes of our modeling, rather than historical occurrences in 2017. The revised sentence now explicitly mentions that the costs are estimated and model-derived, and it also reflects the shift in our reporting approach to focus on the magnitude of sectoral contributions rather than marginal benefits. The updated sentence is as follows:

“The model estimates that in 2017, the integrated costs for seven subsectors exceeded 100 billion dollars. The highest integrated costs were associated with bituminous coal combustion in the energy generation sector, followed by industrial consumption of bituminous coal, diesel-powered vehicles, hydraulic cement production, iron and steel production, domestic consumption of bituminous coal, and agricultural fertilizer application and livestock management (**Table S1**).”

Comment 17

Line 217 – start a new section and use the header from line 172. Breaking up these sections will clarify that a new analysis is starting.

[Response]

In response to your valuable suggestion, we have initiated a new section starting from line 217. This reorganization delineates the start of a new analysis, thereby improving the overall flow of the manuscript. Thank you for guiding this improvement. In addition, we have revised the header from line 172 because the revised analysis focuses on the magnitude of sectoral and spatial contribution to social costs instead of the marginal costs. The revised header is as follows:

“Integrated costs from climate change and health damage modify sectoral and spatial control priorities”.

Comment 18

Line 217 – First sentence should be clearer about what you mean by integrated benefits.

Are you summing the two? Add a reference to Figure 3a.

[Response]

We have amended the sentence in question to enhance clarity regarding the term “integrated benefits.” We now explicitly define integrated benefits as the sum of social benefits from CO₂-related climate change and PM_{2.5} exposure-related health damage as follows.

“Hence, we suggest employing a unified indicator, calculated as the sum of social costs from CO₂-related climate change and PM_{2.5} exposure-related health damage. This indicator is proposed for harmonizing sectoral and spatial control objectives in co-optimal mitigation strategies.”

Figure 3a has been separated and revised as an individual figure as suggested by Reviewer 2. We have added a reference to the figure in the following sentences.

“Spatially, the distributions of social costs from health damage and climate change, as well as the integrated costs, were uneven (**Figure 3 a-c**). The distributions of the contributions of climate change to social costs are skewed toward less populated regions, while the distributions of social costs from health damage are skewed toward densely populated regions (**Figure 3d**).”

Comment 19

Line 267 – Great note about the importance of the atmospheric conditions being an important consideration in this analysis.

[Response]

Thank you for your positive feedback on our emphasis on atmospheric conditions in the analysis. We have moved the sentences to the “Discussion” section (lines 328-332 in the revised manuscript) following your suggestions in comment 21. The sentences now read as follow.

“The Sichuan Basin, for instance, is highlighted as an important contributor to health damage but not to CO₂ emissions, illustrating the nuanced insights provided by adjoint-based source attribution, which accounts for the complex interplay of factors, including population density and terrain, which can exacerbate PM_{2.5} accumulation and secondary aerosol formation⁴³.”

Comment 20

Figure 3. Remove the “_” from the labels.

The lines connecting the cities could be thicker and I can’t see the difference between black and blue. The lines could all be the same color, it doesn’t add much depth to the figure analysis for them to be different.

[Response]

Thank you for your detailed suggestions aimed at enhancing the figure. We have

removed the “_” from the labels and increased the line thickness connecting the provinces in Figure 3b, as recommended. Additionally, we have separated the two subplots in Figure 3 into distinct figures to facilitate easier interpretation following the suggestion from Reviewer 2.

Figure 4. Sectoral contributions to social cost from CO₂-related climate change and integrated cost in each province in mainland China. The provinces were ranked in descending order based on their contributions to the social cost of CO₂-related climate change (left column) and the integrated cost (right column). The sectoral contributions in each province are plotted in the bar chart. The provinces are color coded based on their predominant subsectors.

Comment 21

Overall, this section reads like a Results and Discussion section. If this is intended to just be Results, a good portion of the information should be moved to another section.

[Response]

Thank you for your insightful suggestion regarding the structure of the section. In response, we have revised the section to clearly delineate results from discussion. In specific, interpretive analyses presented in lines 200-216, 265-270, and 282-288 are relocated to the discussion section. The revised discussion section now reads:

“The substantial contribution of monetized health damage to integrated costs suggests that sectoral control priorities for addressing air pollution may not align with the

primary targets of decarbonization plans. Notably, the combustion of bituminous coal for energy generation significantly contributes to the social costs of CO₂-related climate change, and its health co-benefits are well studied^{16,34}. However, in terms of health damage attributable to PM_{2.5} exposure, its contribution is comparatively less than that of industrial and domestic bituminous coal combustion and diesel vehicle emissions. Our assessment indicates that phasing out existing coal-fired power plants in mainland China in 2017 would reduce national losses from health damage by only 5% due to their minor role in reducing air pollutant emissions. Moreover, the health co-benefits of decarbonizing large-scale boilers could diminish if coal-fired power plants extensively adopt ultralow emission control technologies³⁵. Scenario analyses based on Nationally Determined Contributions (NDCs) also reveal that climate policies alone are insufficient to achieve the air quality standards necessary for public health protection^{11,14}. Conversely, despite a minimal contribution to CO₂ emissions, fuel switches in the domestic sector are prioritized when health damage is integrated into the social cost assessment. The social cost of health damage significantly contributes to the integrated costs in regions such as Beijing, Hebei, and Shanxi. The 2017 “Clean Heating Plan,” which targets coal-fired household stoves in the “2 + 26” cities, has reduced domestic coal combustion in these areas³⁶. Domestic solid fuel combustion also contributes more than 20% of the total integrated costs in northern provinces such as Jilin, Heilongjiang, and Gansu, justifying expanded residential fuel switch programs in less-developed northern regions.

A comparison of integrated costs at 36 km by 36 km resolution facilitates the development of nuanced decarbonization strategies tailored to regional, provincial, and city levels. The spatial control priorities based on integrated costs differ from those guided solely by monetized climate impacts. Unlike long-term greenhouse gases, PM_{2.5} and its precursors, which have shorter lifetimes, significantly impact health depending on the proximity of the source³⁷. Including monetized health damage in integrated costs highlights the need to focus on emission control in densely populated areas such as Shanghai, Chongqing, Zhengzhou, and the BHT city cluster. Urbanization offers opportunities for climate change mitigation through improved energy efficiency and public transportation³⁸ but also concentrates emissions, increasing air pollution-related health risks in cities³⁹. A lower dependency of health damage on population density in cities with minimal domestic solid fuel consumption endorses initiatives such as the clean heating campaign to mitigate urbanization's impact on air quality. These findings support the integration of the Low-Carbon Cities Program into China's NDC and its global application^{40,41}, emphasizing population density in pilot program selection for enhanced health co-benefits. Neglecting health impacts in mitigation investments, such as carbon trading, could lead to prioritizing CO₂ emission reductions in less-developed western regions because reducing emissions is more cost-effective in regions with outdated combustion and control technologies⁴².

The sectoral and spatial disparities in the source contributions to health damage and CO₂ emissions identified in our study underscore the necessity for integrated assessments at a high spatial resolution to inform balanced strategies that address both

air pollution and climate change mitigation. CMAQ Adjoint's ability to assess the marginal benefits of emission reductions on a granular scale enables high-resolution source attribution, accounting for emission profiles, population density, and atmospheric conditions. The Sichuan Basin, for instance, is highlighted as an important contributor to health damage but not to CO₂ emissions, illustrating the nuanced insights provided by adjoint-based source attribution, which accounts for the complex interplay of factors, including population density and terrain, which can exacerbate PM_{2.5} accumulation and secondary aerosol formation⁴³.

Our results are subject to several limitations and uncertainties. First, limited by the availability of up-to-date high-resolution emission inventories with detailed source information, our source attribution analysis is based on adjoint sensitivities simulated for the year 2017. Considering China's recent advancements in air pollution control, such as the adoption of ultra-low emission standards in power plants and the clean heating campaign, changes in the emission profile could alter the source attribution results. Second, our health impact assessment focuses exclusively on premature deaths due to ambient PM_{2.5} exposure, and excludes indoor exposures. The contribution of emissions from domestic solid fuel combustion to health damage is likely to be more pronounced when considering indoor exposure, given that both emission and exposure predominantly occur within indoor environments⁴⁴⁻⁴⁶. Third, the integrated cost assessment is sensitive to the monetized values of health damage and climate impacts. There is considerable variability in both the VSL and the SCC estimates, spanning several orders of magnitude. The global SCC used in this study is greater than the country-level SCC, and reflects the worldwide impact of CO₂ emissions⁴⁷. This implies that the relative importance of health damage, and thus the focus on mitigating pollution from heavily polluted sources, would be more pronounced when considering local social costs. Future studies constraining the monetized estimates of VSL and SCC are fundamental to ensuring robust and informed policymaking.”

Comment 22**Discussion and Conclusion**

Lines 294 – 295 How do your findings inform localized decarbonization plans? At what jurisdictional level is “local”?

[Response]

In our study, “local” pertains to both city and provincial levels. The detailed analysis of sectoral contributions to integrated benefits, conducted at a high resolution of 36 km by 36 km, is instrumental in guiding mitigation plans at both provincial and city levels. Policymakers can leverage these findings to pinpoint and prioritize subsectors that would yield the greatest integrated benefits within their coordinated control strategies. The sentence is revised for clarification as follows,

“A comparison of integrated costs at 36 km by 36 km resolution facilitates the development of nuanced decarbonization strategies tailored to regional, provincial, and city levels.”

Comment 23

Line 309 – what do you mean by addressed?

[Response]

By “addressed” we mean that the importance of emission mitigation in these subsectors are emphasized. We have removed the sentence originally in line 309, as our revised analysis no longer centers on MHB and its implications. This change aligns with the recommendations from both you and Reviewer 2. In its place, the revised text includes an in-depth discussion on the differing contributions of coal-fired energy generation to CO₂ emissions and health damage, reflecting a more targeted analysis in line with our research scope.

“Notably, the combustion of bituminous coal for energy generation significantly contributes to the social costs of CO₂-related climate change, and its health co-benefits are well studied^{16,34}. However, in terms of health damage attributable to PM_{2.5} exposure, its contribution is comparatively less than that of industrial and domestic bituminous coal combustion and diesel vehicle emissions.”

Comment 24

Line 312 - Add results and citations to studies that consider the health impact of indoor domestic pollution in China.

[Response]

Studies have demonstrated that indoor PM_{2.5} significantly contributes to the overall PM_{2.5} exposure and its associated health impacts in China (Yun, et al., 2020; Zhao, et al., 2018). In light of your comment, we have updated line 312 to incorporate these findings. Additionally, following your suggestion in comment 21, this revised sentence has been moved to the discussion section.

“The contribution of emissions from domestic solid fuel combustion to health damage is likely to be more pronounced when considering indoor exposure, given that both emission and exposure predominantly occur within indoor environments⁴⁴⁻⁴⁶.”

References

45. Yun, X.; Shen, G.; Shen, H.; Meng, W.; Chen, Y.; Xu, H.; Ren, Y.; Zhong, Q.; Du, W.; Ma, J.; Cheng, H.; Wang, X.; Liu, J.; Wang, X.; Li, B.; Hu, J.; Wan, Y.; Shu, T., Residential solid fuel emissions contribute significantly to air pollution and associated health impacts in China. *Sci. Adv.* **2020**, *6*, (44), doi:10.1126/sciadv.aba7621.
46. Zhao, B.; Zheng, H.; Wang, S.; Smith, K. R.; Lu, X.; Aunan, K.; Gu, Y.; Wang, Y.; Ding, D.; Xing, J.; Fu, X.; Yang, X.; Liou, K.; Hao, J., Change in household fuels dominates the decrease in PM_{2.5} exposure and premature mortality in China in 2005–2015. *Proc. Natl. Acad. Sci.* **2018**, *115*, (49), 12401-12406.

Comment 25

Much of what is included in the Results should be moved to the Discussion. That will help flesh out this section.

[Response]

The interpretive analyses detailed in lines 200-216, 265-270, and 282-288 have been moved to the discussion section in accordance with the suggestion in comment 21. For the revised discussion, please see our response to comment 21.

Comment 26

Other components to consider: who is the integrated analysis intended for? Policymakers, city officials, etc.

[Response]

Thank you for raising the question on our study's application. The integrated analysis is primarily designed for policymakers and environmental regulators at various levels, including national, provincial, and city officials. A key message to the policymakers at national level is that sectoral control priorities to address air pollution may not align with the primary targets of decarbonization plans. The detailed analysis of sectoral contributions to integrated benefits, conducted at a high resolution of 36 km by 36 km, is instrumental in guiding mitigation plans at national, as well as provincial and city levels. Provincial and city level policymakers can leverage these findings to pinpoint and prioritize subsectors that would yield the greatest integrated benefits within their coordinated control strategies. The following sentences are added in the revised "Discussion" section to illustrate our study's application.

Line 305-306 in the revised manuscript: "A comparison of integrated costs at 36 km by 36 km resolution facilitates the development of nuanced decarbonization strategies tailored to regional, provincial, and city levels."

Comment 27

There should justification for this use of the CMAQ model as well as discussion of the model and integrated benefits methods limitations. Not including indoor air pollution exposure would be a part of that.

[Response]

Thank you again for the valuable feedback. The Community Multiscale Air Quality (CMAQ) model was selected for its robust validation and broad acceptance in air quality and source impact assessments (Lu et al., 2010; Zhang et al., 2019). The model's effectiveness is demonstrated in our study through its satisfactory performance in predicting ambient PM_{2.5} concentrations in China, a crucial aspect for accurately attributing PM_{2.5}-related health impacts. Additionally, the advanced multiphase adjoint model of CMAQ (CMAQ-Adjoint) enhances our ability to assess emission source impacts on PM_{2.5} and associated health consequences, offering advantages over reduced form models. To further justify our choice of CMAQ and CMAQ Adjoint, we have

included references to previous studies utilizing these models for evaluating PM_{2.5} concentration and source impacts in line 351. The strengths of CMAQ Adjoint in high-resolution integrated source contribution assessment and its capacity to delineate source attribution under complex atmospheric conditions are further discussed in the revised "Discussion" section.

Line 351 (line 358-361 in the revised manuscript): “CMAQ has been widely applied and validated for simulating ambient PM_{2.5} concentrations and assessing source impacts across various regions, including China ^{49,50}. The CMAQ Adjoint has also been successfully applied in source attribution for air pollution-related health impacts and inverse modeling ^{51,52}”

Line 323-332 in the revised manuscript: “The sectoral and spatial disparities in the source contributions to health damage and CO₂ emissions identified in our study underscore the necessity for integrated assessments at a high spatial resolution to inform balanced strategies that address both air pollution and climate change mitigation. CMAQ Adjoint's ability to assess the marginal benefits of emission reductions on a granular scale enables high-resolution source attribution, accounting for emission profiles, population density, and atmospheric conditions. The Sichuan Basin, for instance, is highlighted as an important contributor to health damage but not to CO₂ emissions, illustrating the nuanced insights provided by adjoint-based source attribution, which accounts for the complex interplay of factors, including population density and terrain, which can exacerbate PM_{2.5} accumulation and secondary aerosol formation ⁴³.”

We recognize the importance of acknowledging the model's limitations to contextualize our findings and conclusions. A detailed discussion regarding the limitations of our modeling approach and the integrated costs methods has been added to the final paragraph of the "Discussion" section.

“Our results are subject to several limitations and uncertainties. First, limited by the availability of up-to-date high-resolution emission inventories with detailed source information, our source attribution analysis is based on adjoint sensitivities simulated for the year 2017. Considering China's recent advancements in air pollution control, such as the adoption of ultra-low emission standards in power plants and the clean heating campaign, changes in the emission profile could alter the source attribution results. Second, our health impact assessment focuses exclusively on premature deaths due to ambient PM_{2.5} exposure, and excludes indoor exposures. The contribution of emissions from domestic solid fuel combustion to health damage is likely to be more pronounced when considering indoor exposure, given that both emission and exposure predominantly occur within indoor environments ⁴⁴⁻⁴⁶. Third, the integrated cost assessment is sensitive to the monetized values of health damage and climate impacts. There is considerable variability in both the VSL and the SCC estimates, spanning several orders of magnitude. The global SCC used in this study is greater than the country-level SCC, and reflects the worldwide impact of CO₂ emissions ⁴⁷. This implies that the relative importance of health damage, and thus the focus on mitigating pollution from heavily polluted sources, would be more pronounced when considering local

social costs. Future studies constraining the monetized estimates of VSL and SCC are fundamental to ensuring robust and informed policymaking.”

References

49. Liu, X.; Zhang, Y.; Xing, J.; Zhang, Q.; Wang, K.; Streets, D. G.; Jang, G.; Wang, W.; Hao, J., Understanding of regional air pollution over China using CMAQ, part II. Process analysis and sensitivity of ozone and particulate matter to precursor emissions. *Atmos. Environ.* **2010**, *44*, (30), 3719-3727.
50. Zhang, Q. et al. Drivers of improved PM_{2.5} air quality in China from 2013 to 2017. *Proc. Natl. Acad. Sci.* **2019**, *116*, 24463-24469.

Reviewer #2 (Remarks to the Author):

Comment 1

Thank you for the opportunity to review this manuscript that estimates the air quality-related health co-benefits of reducing CO₂ emissions in mainland China across 35 subsectors, using CMAQ Adjoint and a high-resolution emissions inventory disaggregated into detailed sectors and fuels.

The modelling appears sound, and the authors are correct that targeting specific sectors and locations for emissions reduction is a more efficient way of improving public health. However, I believe that the results as they stand are very difficult to interpret, in large part because they are expressed in terms of "HB", which represents the air quality-driven health burden of each source divided by the CO₂ emissions from that source.

[Response]

Thank you for your insightful feedback and valuable comments on our manuscript. We appreciate your recognition of the soundness of our modeling approach and the relevance of our findings in the context of targeted emission reduction for greater health co-benefits. We recognize your concerns regarding the clarity and interpretability of our results, especially those presented using the "HB" metric. We concur with your observation that the "HB" metric may not provide an intuitive framework for evaluating the health co-benefits associated with sectoral or regional emissions reductions. In response to your constructive critique, we have undertaken a comprehensive revision of our analytical framework. The revised analysis transitions from employing the "HB" metric to a more detailed examination of the variations in source-specific and regional contributions to CO₂ emissions and their associated health impacts. We believe these modifications have enhanced the clarity of our findings. The details of these revisions have been thoroughly outlined in the point-by-point responses to your comments, as presented below.

Comment 2

The authors seem to be suggesting that, so long as we are agnostic as to which ton of CO₂ emissions to abate, we might as well do this from a source and location where there are higher health benefits. If this was the right analytical frame, then HB would be a meaningful unit to report. This frame is unwarranted for several reasons: (1) we are not agnostic as to which ton of CO₂ emissions to abate; some are far more costly to abate than others (both in terms of implementing the abatement technology and the potential aggregate effects on economic output); (2) even if we were agnostic as to which ton of CO₂ emissions to abate, it is not clear that the most straightforward way of doing so would reduce the air pollution health burden by the same amount. For example, CCS on coal plants will reduce CO₂ but leave air quality unchanged; SO₂ scrubbers will improve air quality but leave CO₂ unchanged. (There are many more such examples across the subsectors listed.)

The claim that for every % reduction in CO₂ emissions there is an equivalent %

reduction in air pollutant emissions across all subsectors and locations is really only warranted in the very particular case of 'phasing out' that pollution source (e.g., avoiding burning fuel). This seems to be the main concern of the paper, leaving out all the other policy options, some of which may be more likely and cost-effective in the short term (and which will, if implemented, completely change the results of this analysis).

The analysis also leaves out "non-synergistic" subsectors---even though these are responsible for over a quarter of the deaths---even though (e.g., in the case of agriculture) these subsectors can be large sources of non-combustion-related, non-CO₂-related greenhouse gas emissions. In all, this paper has a somewhat narrow conception of air quality and climate mitigation options (only being about phasing out fuel combustion), and a very simplistic view of how those would be implemented in policy (no consideration of economic costs/benefits, and only concerned with the sources to target rather than the actual mitigation options).

To overcome these issues, I recommend that the authors (a) directly report the health damage alongside the CO₂ emissions, rather than the HB metric; (b) expand their analysis to include "non-synergistic sectors" and more greenhouse gas emissions (not just CO₂), or make it clear from the outset that they are only considering more traditional sources of air pollution (e.g., fuel combustion in industry/transport); (c) add some more context related to the economic costs and benefits, if not of mitigation options then at least of sectoral output (e.g., how much does each sector currently contribute to GDP?).

[Response]

We are grateful for your comprehensive critique and insightful observations regarding our analytical approach. Your feedback provides a valuable opportunity for us to refine our methodology and clarify our results and findings. Our study is motivated by the fact that the measures to mitigate air pollution are evolving from traditional end-of-pipe controls towards more integrated approaches of low-carbon development, including methods such as low-carbon energy production, enhancement of energy efficiency, and industrial restructuring. This shift is driven by the recognition that the potential for further reductions in air pollution through traditional end-of-pipe controls is becoming increasingly constrained, as these measures have already achieved widespread implementation (Xing et al., 2020, UNFCCC, 2021).

The primary aim of our study is to investigate the alignment of decarbonization efforts with the significant sources and regional contributors to air pollution-related health damage. In response to your suggestion, we have expanded our analysis to encompass non-synergistic sectors that have substantial contribution to ambient PM_{2.5} exposure and its associated health impacts. Furthermore, we concur with your comment that the HB metric does not adequately capture the relationship between CO₂ emission reductions and air quality improvement, given that our analysis does not evaluate the co-benefits specific to mitigation measures. In accordance to your suggestions, we have revised our analytical framework and focused on the disparities between source contributions to CO₂ emissions and health damage. Our revised results quantitatively

show that the primary sectors targeted for decarbonization do not necessarily align with the principal contributors to air pollution-related health damage. Additionally, our high-resolution integrated costs analysis for detailed subsectors can help identify key subsectors and regions to inform more effective synergistic control.

To clearly convey the objective of our study and the subsectors included in our revised analysis, we have rewritten the concluding paragraph of the Introduction.

The rewritten last paragraph of the “Introduction” section now reads:

“In this study, we perform a detailed quantitative analysis to delineate the contributions of diverse sectors and their associated fuel/process combinations to CO₂ emissions and the health damage related to air pollution in China. This analysis encompasses critical sectors such as energy production, industry, transportation, domestic activities (both residential and commercial sources), and agriculture. The health impact assessment focuses on nationwide premature mortality attributable to ambient PM_{2.5} exposure. Our coordinated source attribution analysis integrates coupled emission inventories for both air pollutants and CO₂ with an advanced adjoint tool developed for a regional air quality model (CMAQ Adjoint). This tool allows for the efficient calculation of backward sensitivities, thereby quantifying the impact of high-resolution emission changes on nationwide premature mortality counts. The analysis includes impact of emission from speciated primary PM_{2.5}, which include organic carbon (OC), elementary carbon (EC), and other primary PM_{2.5} particles, as well as from principal precursors of secondary PM_{2.5}, which include sulfur dioxide (SO₂), nitrogen oxides (NO_x), and ammonia (NH₃). Moreover, we propose using the integrated costs – representing the aggregate of monetized social costs from health damage and climate change – as a comprehensive metric for prioritizing sectoral and spatial control targets.”

The modifications made in response to your three key suggestions are detailed below.

(a) directly report the health damage alongside the CO₂ emissions, rather than the HB metric

We have revised our manuscript and figures to report the magnitude of sectoral and spatial contributions to health damage and CO₂ emissions.

Firstly, we have modified Figure 1 to directly compare the source contribution to premature deaths and CO₂ emissions from each subsector in panel c. Additionally, Figure S2 has been updated to align with these updated results. The section spanning lines 118-131 has been thoroughly rewritten to incorporate discussions regarding the extent of source contributions to health damage and CO₂ emissions.

“Seven subsectors were identified as significant contributors to health damage, each of which was attributable for more than 100,000 premature deaths (**Figure 1c**). These subsectors include bituminous coal combustion in the domestic, energy generation, and industrial sectors; emissions from diesel-powered vehicles; and activities in hydraulic cement production, iron and steel production, as well as agricultural fertilizer application and livestock management. While these subsectors comparably contribute

to health damage, their contributions to CO₂ emissions differ substantially. For instance, agricultural activities significantly contribute to PM_{2.5}-related health damage from NH₃ emissions without being major contributors to CO₂ emissions directly. This discrepancy is also observed in the analysis of CO₂ emissions from other subsectors. For example, the CO₂ emissions from domestic bituminous coal combustion are 52-93% lower than those from the remaining five subsectors. In terms of the speciated emission contribution to health damage, more than half of the health damage from hydraulic cement and iron production, as well as domestic bituminous coal combustion, is attributed to primary PM_{2.5} emissions. In contrast, health damage from bituminous coal combustion in the energy generation and industrial sectors, as well as from diesel vehicles, is predominantly attributable to SO₂ and NO_x emissions (**Figure S2**). The impact of various fuel types and industrial processes on premature deaths within a sector also shows substantial disparities. For example, diesel-powered vehicles, which are responsible for 210,000 premature deaths, contribute eightfold more to health damage than do gasoline-powered vehicles, which emphasizes the great importance of diesel vehicle electrification in reducing air pollution and associated health risks. Overall, the ratio of source contributions to health damage versus to CO₂ emissions varies extensively across subsectors, ranging from 0.094 in the industrial combustion of dry natural gas to 22 in the domestic combustion of unorganized waste. This noteworthy variation in source profiles for health damage and CO₂ emissions underscores that strategies aimed at reducing air pollution-related health damage may not always align with those targeting decarbonization. This necessitates the formulation of coordinated, synergistic control plans to effectively balance the two objectives.”

Secondly, in lieu of the previously utilized HB metric, we have adopted a new metric that evaluates the ratio of each source's percentage contribution to nationwide PM_{2.5}-related health damage versus its contribution to CO₂ emissions. This adjustment shifts the metric from quantifying marginal health co-benefits of CO₂ emission reduction to directly highlighting disparities in source contributions. Accommodating this shift, we have made the following revisions.

Figure 2a and Figure 2c have been updated, and corresponding text has been comprehensively rewritten to reflect these revisions.

The sentences in lines 134-142 are rewritten as follows.

“The integration of high-resolution health damage attribution data with the coordinate CO₂ emission inventory also enables the delineation of spatial heterogeneities in contributions to health damage and CO₂ emissions. By calculating the ratio of the percentage contributions to the nationwide PM_{2.5}-related health damage versus to CO₂ emissions at the gridded or regional levels, we can assess the relative significance of the contributions. In densely populated areas (**Figure S3**), including the eastern and central regions and the Sichuan Basin, the contribution to health damage notably surpassed that to CO₂ emissions (**Figure 2a**). The highest ratio was observed in Hubei (3.9), followed by Henan and Chongqing (3.8 and 3.7, respectively), whereas the lowest ratio was recorded in Xizang (0.11).”

The sentences in lines 150-152 have been revised as follows.

“The spatial distribution of the health damage to CO₂ emissions ratio is significantly influenced by population density. A log-linear regression analysis at the city level shows positive correlation between population density and the ratio of each city’s percentage contribution to nationwide PM_{2.5}-related health damage versus its contribution to CO₂ emissions (**Figure 2c**).”

The sentences in lines 155-170 have been revised as follows.

“Additionally, the source profile, particularly the domestic energy structure, contributes to the spatial variations in the ratio. The color-coded scatter plot in **Figure 2c** illustrates the impact of reliance on solid fuels for domestic energy needs on the correlation between the ratio of contributions to health damage versus that to CO₂ emissions and population density. Notably, this correlation intensifies in cities with a greater dependence on solid fuels for domestic energy, explaining more than half of the variance in city-level ratios. Specifically, a 1% increase in population density correlates with a 0.47% increase in the ratio. Conversely, in cities where domestic solid fuel consumption constitutes less than 1% of total provincial energy usage, population density accounts for merely 4% of the ratio variance, with a 1% population increase leading to a modest 0.12% increase in the ratio. The higher ratio in Beijing than in other megacities, including Shanghai, Guangzhou, and Shenzhen, further demonstrates the critical role of domestic solid fuel consumption (**Figure S5**). Although Beijing has already decommissioned its energy-intensive industries and coal-fired power plants, it is the only city among the four megacities whose contribution to health damage exceeds its contribution to CO₂ emissions. This increase in Beijing’s contribution to health damage was primarily attributed to the consumption of 1.8 Mt of raw coal by rural residents in 2017²⁸.”

We have also revised Figure 2b to directly compare percentage contributions to nationwide premature deaths and CO₂ emissions. The sentences in lines 142-149 have been revised based on the updated results.

“Spatial heterogeneity in the source contributions to health damage and CO₂ emissions is also evident within individual sectors. For instance, in the energy generation sector, Inner Mongolia contributes marginally more to CO₂ emissions (3.8%) than Shandong (3.5%), yet its impact on health damage (0.75%) is significantly lower than Shandong's (1.3%) (**Figure 2b**). Similarly, in the industrial sector, Guangdong's CO₂ emissions surpass those of Anhui and Hebei (18%), yet its health damage contribution is merely half of that observed in the latter provinces. This spatial variability is influenced not only by population density but also by the intensity of air pollutant emissions (**Figure S4**). For example, Sichuan, with its lower coal quality leading to higher emissions of PM_{2.5} and SO₂ per ton of CO₂ emission, exhibits the most pronounced disparity in the energy generation sector. Similarly, the elevated PM_{2.5} emissions associated with clinker production in the hydraulic cement manufacturing make Anhui and Hebei the regions with the highest discrepancy in the industrial sector.”

Thirdly, the discussion regarding monetized health co-benefits (MHB) has been updated to directly quantify monetized health damage. The monetized health damage are compared with monetized climate impacts in the revised Figure 1c. Figure S6 has been deleted because it contains repetitive information with the revised Figure 1c. Corresponding text in lines 174-199 has been updated as follows:

“We further monetized health damage and climate impacts by applying a uniform value of statistical life (VSL) of 1.33 million USD per statistical death and a direct social cost of carbon (SCC) of 100 USD per ton of CO₂ emission (**Methods**). The monetized costs for health damage and climate impacts attributable to different subsectors are depicted along the minor axis in **Figure 1c**. For the 36 synergistic subsectors, the monetized health damage and CO₂-related climate impacts fall along the one-by-one line on a logarithmic scale, with the monetized health damage exceeding the corresponding monetized climate impacts for half of the subsectors. This finding indicates that the social costs due to climate change are comparable to the monetized concurrent health losses attributable to air pollution. The VSL value employed in this study, following the Organisation for Economic Co-operation and Development (OECD) recommended benchmark, is 2–3 times lower than the estimates recommended by the United States Environmental Protection Agency (USEPA). If higher VSL estimates were used (**Figure S6**), monetized health damage would surpass climate impacts for the majority of subsectors. This alignment, or in certain cases, the excess of monetized health damage related to climate impacts, emphasizes the near-term health benefits as an important incentive for CO₂ emission reduction ^{12,29} and justifies more ambitious decarbonization plans.”

Moreover, instead of the integrated benefits at 1% reduction of CO₂ emissions, we propose to use integrated cost as the sum of monetized health damage and climate impacts as the unified indicator for coordinating sectoral and spatial control objectives for co-optimal mitigation plans. The following revisions were made based on the revised indicator.

Line 217-227: “Hence, we suggest employing a unified indicator, calculated as the sum of social costs from CO₂-related climate change and PM_{2.5} exposure-related health damage. This indicator is proposed for harmonizing sectoral and spatial control objectives in co-optimal mitigation strategies. The model estimates that in 2017, the integrated costs for seven subsectors exceeded 100 billion dollars. The highest integrated costs were associated with bituminous coal combustion in the energy generation sector, followed by industrial consumption of bituminous coal, diesel-powered vehicles, hydraulic cement production, iron and steel production, domestic consumption of bituminous coal, and agricultural fertilizer application and livestock management (**Table S1**). Except for the last non-synergistic subsector, the other six subsectors all heavily relied on coal or diesel fuel.”

Lastly, in accordance with the revised analytical framework, the “Health Co-benefit Evaluation” section within the “Methods” has been updated to detail the health impact assessment and integrated cost analysis as follows.

Line 398-425: “We quantify the health damage attributable to major anthropogenic sectors and their subsectors at a monthly resolution by combining the sectoral air pollution emission amount and the adjoint sensitivity as follows (eq. 2):

$$M_{g,m,s} = \sum_p \frac{\partial J}{\partial E_{p,g,m}} \times \Delta E_{p,g,m,s}, \text{ eq. 2}$$

where $\frac{\partial J}{\partial E_{p,g,m}}$ represents the marginal change in the annual premature death attributable to ambient PM_{2.5} exposure in China in 2017 per unit change in emissions of air pollutant p in grid g and month m . The total health damage from sector s equals the sum of premature deaths attributable to the emissions of each primary PM_{2.5} species and its precursors, including OC, EC, other primary PM_{2.5}, SO₂, NO_x, and NH₃. The contributions from VOC emissions were also not discussed because of their minor contributions to PM_{2.5}-related health damage (**Figure S10**). The monthly averaged adjoint sensitivities for each air pollutant species ($\frac{\partial J}{\partial E_{p,g,m}}$) were calculated as emission-weighted averages from the three-dimensional daily adjoint sensitivity outputs from the CMAQ adjoint.”

Line 426-427:

“Monetized assessment and integrated costs

The social cost of PM_{2.5} exposure-related health damage can be evaluated as the monetized value of premature deaths based on the economic VSL.”

Line 439-440: “The corresponding social cost of CO₂-related climate change for each subsector can be evaluated based on the CO₂ emission amount and SCC.”

The following sentences have been added in line 444: “The SCC estimate is close to a recent expert elicitation estimate (approximately 80–100 USD when outliers were trimmed)⁶⁵. We further evaluate the integrated costs as the sum of the social costs of health damage and climate change.”

The sentences in line 436-438 and text in supplementary methods 1 have been combined to detailed the uncertainty analysis for monetized assessment and added to the end of the “Monetized assessment and integrated costs” section as follows:

“The impacts of VSL and SCC uncertainties on the monetized assessment are evaluated by perturbing the adopted values. A high estimate of VSL was calculated using the baseline VSL suggested by the U.S. Environmental Protection Agency (USEPA), the average GDP per capita for the U.S., and an income elasticity of 0.5⁶⁶. The VSL was calculated to be 4.36 million USD in 2011. Another VSL estimate based on a contingent valuation study in six representative cities in China (0.66 million in 2011 dollars) is viewed as a low estimate of the VSL⁶⁷. The SCC is perturbed from 10 to 1000 USD per ton of CO₂ emission⁶⁴ for uncertainty assessment.”

References

Xing, J.; Lu, X.; Wang, S.; Wang, T.; Ding, D.; Yu, S.; Shindell, D.; Ou, Y.; Morawska, L.; Li, S.; Ren, L.; Zhang, Y.; Loughlin, D.; Zheng, H.; Zhao, B.; Liu, S.; Smith, K. R.; Hao, J., The quest for improved air quality may push China to continue its CO₂ reduction beyond the Paris Commitment. *Proc. Natl. Acad. Sci.* **2020**, *117*, (47), 29535-29542.

UNFCCC NDC Report, China's achievements, new goals and new measures for Nationally Determined Contributions. **2021**. https://unfccc.int/NDCREG?gclid=CjwKCAjw9pGjBhB-EiwAa5jl3PyTftOsvBSjtxajragqACUrSQtJ7uONh3B68_PQfvoeowatsOZfCxoCaVkQAvD_BwE (Assessed on 12/10/2022).

Figure 1. Disparities in source attributions of CO₂ emissions and health damage. (a) Source contributions to CO₂ emissions in China in 2017 based on the coordinated CO₂ emission inventory with the air pollutants. The lower bar represents the contributions of different sectors, and the upper bar represents the contributions of different fuel/process categories. **(b)** Source contributions to premature deaths attributable to long-term ambient PM_{2.5} exposure in China in 2017 based on the CMAQ adjoint simulation. The left bar represents the contributions of different sectors, and the right bar represents the contributions of different fuel/process categories. The shaded areas are non-synergistic subsectors without direct CO₂ emissions. **(c)** Comparison of impacts on health damage and CO₂ emissions for each subsector. The corresponding monetized impacts are also shown on the minor axis based on a uniform statistical life value (1.33 million USD per premature death) and social cost of carbon (100 USD per ton of CO₂ emission). The abbreviations for the fuel/process categories are listed in **Table S2**.

Figure S2 Speciated contribution to PM_{2.5} exposure-related premature deaths and CO₂ emissions from top contributors to premature deaths. The chart shows source contributions to PM_{2.5} exposure-related premature deaths and CO₂ emissions from the seven subsectors with the highest contribution to premature deaths, including electric bituminous coal combustion, industrial bituminous coal combustion, hydraulic cement production, iron and steel production, diesel vehicles, domestic bituminous coal combustion, and fertilizer application and livestock management. The stacked bars show the speciated contributions from emissions of primary PM_{2.5} and gas precursors.

Figure 2. Disparities in spatial distributions of CO₂ emissions and health damage. (a) Ratio between the gridded contribution to PM_{2.5} exposure-related health damage and to CO₂ emissions. The provincial boundary shapefile is obtained from Harvard Dataverse (<https://doi.org/10.7910/DVN/DBJ3BX>) and is publicly available under the Creative Commons CC0 Public Domain Dedication. (b) Comparison of provincial contributions to health damage and CO₂ emissions in each sector. The red dots represent the ratio of provincial contributions to health damage versus to CO₂ emissions. The provinces are ranked from large to small in population size. The abbreviations of the provinces are listed in **Table S3**. (c) Correlation between population density and the ratio of each city's contributions to health damage versus to CO₂ emissions. The dot color represents the domestic solid fuel usage fraction, which is defined as the fraction of energy consumed as solid fuels in the domestic sector out of the total province-level energy consumption. The solid line and dashed line represent the correlations for cities with domestic solid fuel consumption above and less than 1%, respectively, of the total provincial energy consumption.

Figure S5 Comparison of city contribution to PM_{2.5}-related health damage and CO₂ emissions from top megacities in China. The stacked bars show source contributions to nationwide health damage (upper bars) and CO₂ emissions (lower bars) in each city. The red dots show the ratio between city contributions to health damage and CO₂ emissions.

(b) expand their analysis to include "non-synergistic sectors" and more greenhouse gas emissions (not just CO₂), or make it clear from the outset that they are only considering more traditional sources of air pollution (e.g., fuel combustion in industry/transport);

We have revisited the analysis to include both synergistic and non-synergistic subsectors that significantly contribute to PM_{2.5} exposure-related health damage. All results and figures have been updated to include contribution from non-synergistic subsectors. However, emissions for non-CO₂ GHGs from other non-synergistic sources are not included due to the lack of coupled emission inventories and their relatively minor impact on ambient PM_{2.5}. All non-synergistic subsectors included are added to the revised Table S1.

The contribution of non-synergistic subsectors to premature deaths has been added in the revised Figure 1. In addition, the sentence in line 113 has been revised to clarify the non-synergistic subsectors as follows. This revision includes an adjustment in the contribution of non-synergistic subsectors due to a correction of a previously identified error in the calculation of emissions and health impacts associated with the iron and steel production process.

“Certain subsectors contribute to health losses without being directly linked to CO₂ emissions from fuel combustion. These include non-combustion industrial processes, agricultural fertilizer application and livestock management, and crop residue burning in the domestic sector (Table S1). While the combustion of crop residues does emit

CO₂, this subsector is considered “carbon neutral” based on the assumption that the released CO₂ is later reabsorbed through subsequent biomass regrowth²³. As in our approach for other fuel types, life cycle emissions from biofuel processing and land use changes are not included^{24,25}. We term these subsectors “non-synergistic subsectors”, which account for 23% of the total premature deaths attributable to the five sectors assessed.”

In addition, our updated analysis indicates that the relationship between the source contribution ratio of health damage to CO₂ emissions and population density is influenced by the varying reliance on solid fuels versus coal consumption for domestic energy demand. This observation underscores the significant impacts of non-synergistic sectors. For a more detailed elaboration, please refer to our response to suggestion (a).

Furthermore, we have revised the ranking based on integrated costs to include the monetized health damage attributed to non-synergistic subsectors. This adjustment is reflected in the updated Figure 4 and the following text revisions.

Lines 234-239: “Because of their substantial contribution to health damage, the ranking position of domestic subsectors reliant on solid fuels increased in the integrated cost-based ranking (**Table S1**). In addition to the domestic bituminous coal combustion, indoor crop residue burning and brush wood burning were positioned 9th and 10th, respectively, out of the 53 subsectors assessed in terms of their contribution to integrated costs. However, decarbonization-oriented mitigation measures may not address these two subsectors due to their negligible contribution to CO₂ emissions.”

Lines 271-282: “In six provinces, namely, Henan, Liaoning, Tianjin, Beijing, Hebei, and Shanxi, the primary subsector contributing to integrated costs diverges from the leading subsector contributing to social costs from CO₂-related climate change. In Henan and Liaoning, the predominant subsector shifts from being bituminous coal combustion in the energy generation sector to being bituminous coal combustion in industrial boilers, and in Tianjin, the predominant sector changes to iron and steel production. Although the energy generation sector is the most studied sector for low-carbon pathways^{16,34}, the combined integrated costs from bituminous coal combustion and five energy-intensive processes (iron and steel, coke, brick, hydraulic cement, and lime production) dominated in 26 out of the 31 provinces. Thus, programs to decarbonize industrial boilers and phase out energy-intensive and highly polluting industrial processes are the keys to optimizing low-carbon development in most provinces. For Beijing, Hebei, and Shanxi, the top contributing subsectors shift to domestic bituminous coal combustion, a shift attributed to its substantial contribution to health damage.”

In accordance, we have updated the “Highly Resolved Emission Inventory” section of “Methods” to include non-synergistic subsectors.

Line 398: The phrase “combustion-related” has been deleted.

Line 400-402: Energy generation, industrial, domestic, transportation, and agricultural activities were divided into 6, 24, 15, 5, and 3 subcategories, respectively.

Line 402: The phrase “combustion source” has been replaced with “sector”.

Table S1 The monetized health damage, CO₂ emissions, integrated benefits, and their associated ranks for all source sectors assessed in this study

sector	fuel/process category	monetized health damage (rank)	monitized climate impacts (rank)	integrated costs (rank)
industry	bituminous coal	2.2×10^{11} (1)	2.6×10^{11} (2)	4.8×10^{11} (2)
	anthracite	9.8×10^9 (22)	8.7×10^9 (13)	1.9×10^{10} (20)
	coking coal	4.9×10^9 (26)	5.3×10^9 (16)	1.0×10^{10} (26)
	gas/diesel	2.6×10^{10} (16)	8.9×10^9 (12)	3.5×10^{10} (15)
	dry natural gas	1.4×10^9 (37)	9.0×10^9 (11)	1.0×10^{10} (24)
	crude oil	3.4×10^9 (31)	9.2×10^9 (10)	1.3×10^{10} (22)
	residue fuel oil	1.1×10^9 (39)	2.6×10^9 (21)	3.6×10^9 (35)
	industrial waste	1.5×10^4 (53)	6.2×10^4 (36)	7.7×10^4 (53)
	primary Al production	1.9×10^9 (33)	2.0×10^9 (22)	3.9×10^9 (34)
	hydraulic cement production	1.8×10^{11} (5)	7.6×10^{10} (4)	2.5×10^{11} (4)
	iron and steel production	1.3×10^{11} (7)	1.2×10^{11} (3)	2.4×10^{11} (5)
	coke production	5.7×10^{10} (10)	3.6×10^7 (7)	9.4×10^{10} (8)
	brick production	2.7×10^{10} (15)	1.2×10^{10} (9)	3.9×10^{10} (13)
	petroleum catalytic cracking	1.6×10^9 (36)	0 (N/A)	1.6×10^9 (38)
	lime production	2.8×10^{10} (14)	0 (N/A)	2.8×10^{10} (18)
	glass production	5.3×10^9 (25)	0 (N/A)	5.3×10^9 (32)
	fertilizer production	1.3×10^{10} (20)	0 (N/A)	1.3×10^{10} (21)
	ferroalloy production	8.9×10^9 (23)	0 (N/A)	8.9×10^9 (29)
	lead production	8.7×10^8 (42)	0 (N/A)	8.7×10^8 (45)
	magnesium production	1.6×10^8 (48)	0 (N/A)	1.6×10^8 (49)
zinc production	1.1×10^9 (38)	0 (N/A)	1.1×10^9 (42)	
ammonia production	1.6×10^9 (35)	0 (N/A)	1.6×10^9 (37)	
nickel and copper production	4.2×10^9 (28)	0 (N/A)	4.2×10^9 (33)	
natural gas production	1.2×10^{10} (21)	0 (N/A)	1.2×10^{10} (23)	
energy generation	bituminous coal	1.8×10^{11} (4)	3.5×10^{11} (1)	5.3×10^{11} (1)
	coking coal	2.6×10^7 (51)	5.1×10^7 (33)	7.6×10^7 (52)
	gas/diesel	6.6×10^7 (50)	1.7×10^8 (31)	2.4×10^8 (48)
	dry natural gas	2.3×10^9 (32)	8.0×10^9 (14)	1.0×10^{10} (25)
	residue fuel oil	3.6×10^8 (46)	1.1×10^9 (26)	1.5×10^9 (39)
	solid biomass	2.1×10^8 (47)	6.5×10^8 (27)	8.6×10^8 (46)

Table S1 The monetized health damage, CO₂ emissions, integrated benefits, and their associated ranks for all source sectors assessed in this study (continued)

sector	fuel/process category	monetized health damage (rank)	monitized climate impacts (rank)	integrated costs (rank)
domestic	bituminous coal	2.1×10^{11} (2)	2.6×10^{10} (8)	2.3×10^{11} (6)
	honeycomb/coal briquettes	2.9×10^{10} (13)	3.7×10^9 (18)	3.2×10^{10} (16)
	indoor firewood burning	5.0×10^{10} (11)	3.0×10^9 (20)	5.3×10^{10} (12)
	non-organized waste burning	9.3×10^8 (41)	2.4×10^7 (34)	9.5×10^8 (43)
	patent fuel	8.8×10^9 (24)	1.2×10^9 (25)	1.0×10^{10} (27)
	gas/diesel	1.8×10^9 (34)	1.4×10^9 (23)	3.2×10^9 (36)
	dry natural gas	1.0×10^9 (40)	2.3×10^8 (30)	1.2×10^9 (41)
	liquid petroleum gas	3.7×10^9 (30)	6.1×10^9 (15)	9.8×10^9 (28)
	gas works gas	8.6×10^8 (43)	4.9×10^8 (28)	1.4×10^9 (40)
	coke oven gas	6.1×10^8 (44)	3.4×10^8 (29)	9.5×10^8 (44)
	biogas	1.4×10^7 (52)	8.3×10^7 (32)	9.6×10^7 (51)
	indoor crop residue burning	6.4×10^{10} (8)	0 (N/A)	6.4×10^{10} (9)
	charcoal	4.2×10^8 (45)	0 (N/A)	4.2×10^8 (47)
	indoor corncob burning	3.8×10^{10} (12)	0 (N/A)	3.8×10^{10} (14)
	indoor brush wood burning	6.2×10^{10} (9)	0 (N/A)	6.2×10^{10} (10)
transportation	gasoline	1.9×10^{10} (19)	3.97×10^8	5.58×10^2
	diesel	2.0×10^{11} (3)	5.36×10^8	2.23×10^3
	biogas	4.4×10^9 (27)	1.26×10^7	4.46×10^1
	aviation	4.1×10^9 (29)	3.24×10^7	6.52×10^1
	shipping	1.1×10^8 (49)	1.23×10^5	9.14×10^{-1}
agriculture	gas/diesel	2.6×10^{10} (17)	4.9×10^9 (17)	3.0×10^{10} (17)
	agriculture waste burning	2.0×10^{10} (18)	0 (N/A)	2.0×10^{10} (19)
	fertilizer application and livestock management	1.4×10^{11} (6)	0 (N/A)	1.4×10^{11} (7)

Figure 4. Sectoral contributions to social cost from CO₂-related climate change and integrated cost in each province in mainland China. The provinces were ranked in descending order based on their contributions to the social cost of CO₂-related climate change (left column) and the integrated cost (right column). The sectoral contributions in each province are plotted in the bar chart. The provinces are color coded based on their predominant subsectors.

(c) add some more context related to the economic costs and benefits, if not of mitigation options then at least of sectoral output (e.g., how much does each sector currently contribute to GDP?)

To establish the link between source contribution to social cost from climate change and health damage with sectoral economic outputs or GDP for the subsectors assessed in our study, we leverage a production-based emission inventory mapped to emissions from 42 economic sectors. This mapping is based on the China multi-regional input-output (MRIO) model table for the year 2017 (Zheng, Scientific Data, 2021). In total, trade-related emissions accounted for 90% of the CO₂ emissions and 68% of the PM_{2.5} exposure-related health damage from all subsectors assessed because emissions from household direct energy use are excluded. On average, the integrated social cost of the 42 economic sectors equals 20% of the total GDP, with the sectoral ratio between integrated social cost and GDP varying from 2.4×10^{-4} to 3.4. There are four sectors of which the integrated social cost exceeds their sectoral GDP, specifically 1) the

processing of petroleum, coking, and nuclear fuel, 2) manufacture of non-metallic mineral products, 3) smelting and processing of metals, and 4) production and distribution of electric power and heat power. The comparison between sectoral integrated costs and economic outputs are discussed in the section of “Monetized social costs of CO₂ emissions and PM_{2.5} exposure-related health damage” as follows in lines 203-212 in the revised manuscript.

“We further conducted a comparison between the monetized health damage and climate impacts with sectoral GDP, assigning production-based emissions to 42 economic sectors by using the China multi-regional input-output model (MRIO) table for 2017³⁰ (Supplementary Methods, **Table S4**). On average, the integrated costs of the 42 economic sectors equal 20% of the total GDP. For most of the economic sectors, the integrated costs are lower than 5% of the sectoral GDP (**Figure S7**). However, in four sectors, the integrated costs exceed the sectoral GDP. These sectors include the production and distribution of electric power and heat power; the manufacture of non-metallic mineral products, the processing of petroleum, coking, and nuclear fuel; and the smelting and processing of metals. This comparison further underscores the critical need for decarbonizing electricity generators and energy-intensive industrial processes.”

Detailed methods to attribute the emissions from 42 economic sectors are provided in the Supporting Materials as follow.

“Due to the difference for sector classification between the PKU-FUEL inventory and the China multi-regional input-output model (MRIO) table¹, we assigned the production-based emissions to the 42 economic sectors (**Table S4**) based on the source mapping process described below. Emissions from the electric sector, agriculture, and transportation sector were directly mapped to the corresponding sectors in the MRIO. Emissions from industrial combustion were attributed to the mining and manufacturing sectors utilizing provincial energy balance sheets² and sectoral energy consumption³ as proxies. Emissions from industrial processes were mapped to the corresponding MRIO sectors based on the products categories. Emissions from the commercial sectors were split from the domestic subsectors based on the provincial energy balance sheets². In the lack of sector-specific energy statistics, the commercial emissions were attributed to the tertiary industry sectors using sectoral monetary outputs in the MRIO table as proxies. Residential emissions cannot “flow” in trade, and were thus excluded in the emission mapping process.”

References

1. Zheng, H. et al., Chinese provincial multi-regional input-output database for 2012, 2015, and 2017. *Sci. Data* **2021**, 8, 244, doi:10.1038/s41597-021-01023-5.
2. National Bureau of Statistics of China (ed.), China energy statistical yearbook 2018. China Statics Press: Beijing, 2018.
3. National Bureau of Statistics of China (ed.), China statistical yearbook 2018. China Statics Press:

Beijing, 2018.

Figure S7 Comparison between integrated costs and sectoral GDP. The color-coded circles represent the sectoral MHB with the grey shade showing uncertainty ranges due to the variance of VSL estimates. The dashed lines also represent the range of SCC estimates. The abbreviations of the 42 economic sectors are listed in **Table S3**.

Comment 3

Some more detailed comments:

"35 emission sources"---do you mean subsectors? "Sources" is vague.

[Response]

Indeed, we refer to 35 subsectors, and we apologize for any confusion this may have caused. In the revised manuscript, we have replaced the term “emission sources” with “subsectors” for greater clarity. Furthermore, the initial count of 35 subsectors has been expanded to 53 to encompass both synergistic and non-synergistic subsectors.

The sentence in line 23 is revised as “Using adjoint emission sensitivity modeling, in this study, PM_{2.5}-related mortality were attributed to 53 sector and fuel/process combinations in 2017 with high spatial resolution, revealing notable discrepancies between their contributions to CO₂ emissions and health damage.”

Comment 4

"For example, 45 following the RCP4.5 scenario (which depicts a moderately warming future), researchers have 46 estimated that CO₂ emission reductions can help prevent 1.3 million premature deaths globally 47 in 2050 by reducing air pollutant emissions simultaneously." What is the citation for this statement? Is this source 10 (Shindell et al., 2018)?

[Response]

The citation for this statement is West et al., 2013. We misplaced the reference to this sentence to the next sentence. We apologized for the confusion. The citations have been updated to correct this mistake.

“For example, researchers have estimated that CO₂ emission reductions following the RCP4.5 scenario (which depicts a moderately warming future) can help prevent 1.3 million premature deaths globally by 2050 by reducing air pollutant emissions simultaneously⁹. This number could double under scenarios in which more stringent CO₂ emission control is applied¹².”

References

9. West, J. J.; Smith, S. J.; Silva, R. A.; Naik, V.; Zhang, Y.; Adelman, Z.; Fry, M. M.; Anenberg, S.; Horowitz, L. W.; Lamarque, J. F., Co-benefits of Global Greenhouse Gas Mitigation for Future Air Quality and Human Health. *Nat. Clim. Chang.* **2013**, *3*, (10), 885-889.
12. Shindell, D.; Faluvegi, G.; Seltzer, K.; Shindell, C., Quantified, localized health benefits of accelerated carbon dioxide emissions reductions. *Nat. Clim. Chang.* **2018**, *8*, (4), 291-295.

Comment 5

- Figs 1 and 3 are very difficult to interpret. There is a lot going on in each of them. I suggest having figures that each present one finding very clearly and convincingly.

[Response]

Thank you for the suggestion. We have revised Figures 1 and 3 to facilitate clearer interpretation.

For Figure 1, we have restructured it in response to reviewers' comments, focusing on the disparities in source contributions between PM_{2.5} exposure-related health damage and CO₂ emissions. The revised Figure 1c now features a scatter plot that directly compares the contributions of all assessed subsectors to health damage and CO₂ emissions, rather than focusing on sectoral differences in health co-benefits per ton of CO₂ emission. To enhance clarity, we have transformed the pie charts in panel a into bar charts so that we can align them along the corresponding axis showing the source contribution to health damage and CO₂ emissions from each subsector in the updated panel c. Each bar chart, along with the scatter plot, has been labeled from a to c, with concise descriptions added to each panel for better understanding. Please refer to the revised Figure 1 in our response to comment 2.

Figure 3 has been divided into two separate figures to enhance clarity. The original Figure 3a has been modified to become the new Figure 3, illustrating the spatial distribution of social cost from CO₂-related climate change and PM_{2.5} exposure-related health damage, as well as the integrated costs. We have also incorporated a Quantile-Quantile plot to highlight the skewed distribution of social costs from health damage and the integrated cost towards populated regions, in contrast to the social costs from CO₂ emissions, which are skewed towards less-populated remote regions. The former Figure 3b is now presented as Figure 4, focusing on the comparison between sectoral contributions to social cost from CO₂-related climate change and integrated cost in each province. Based on the updated figures, following revisions have been made in the text. The revised Figure 3 is attached below, and the revised Figure 4 is attached to the response to comment 2.

Line 240-244 and Line 248-250 are revised and combined: “Spatially, the distributions of social costs from health damage and climate change, as well as the integrated costs, were uneven (**Figure 3 a-c**). The distributions of the contributions of climate change to social costs are skewed toward less populated regions, while the distributions of social costs from health damage are skewed toward densely populated regions (**Figure 3d**). The incorporation of health co-benefits amplified the contribution of high-population regions to integrated benefits because of the proximity of air pollution and exposed populations. The areas with high integrated costs were concentrated in populated areas where greater amounts of air pollutants and CO₂ emissions were emitted from fuel combustion to support intensive human activities. The grids that contributed more than 1 billion dollars in integrated costs covered only 11% of the land but hosted 60% of the population of China.”

Lines 251-252: “Consequently, the spatial priorities for synergistic control identified based on the integrated costs differ from those based solely on the social cost of CO₂-related climate change.”

Figure 3. Spatial distribution of social costs from PM_{2.5} exposure-related health damage, CO₂-related climate change, and integrated costs. (a) Spatial distribution of PM_{2.5} exposure-related health damage in China at a 36 km by 36 km resolution in 2017. **(b)** Spatial distribution of social costs from CO₂-related climate change. **(c)** Spatial distribution of the integrated costs, which equal the sum of the monetized health damage and social cost from CO₂-related climate change. **(d)** Quantile–quantile plot of the social cost distribution against the population distribution. The three lines represent comparisons of social costs from CO₂-related climate change, PM_{2.5}-related health damage, and integrated costs.

REVIEWERS' COMMENTS

Reviewer #2 (Remarks to the Author):

The authors have done a satisfactory job of addressing the comments. I recommend publication.

Nature Communications manuscript NCOMMS-23-32437A

Title: Substantial differences in source contributions to carbon emissions and health damage necessitate balanced synergistic control plans in China

REVIEWER COMMENTS

Reviewer #2 (Remarks to the Author):

Comment 1

The authors have done a satisfactory job of addressing the comments. I recommend publication.

[Response]

Thank you again for taking the time to review our paper carefully and for providing us a set of very useful comments and suggestions.